# INFONET: AN EFFICIENT FEED-FORWARD NEURAL ESTIMATOR FOR MUTUAL INFORMATION

## ABSTRACT

Estimating mutual correlations between random variables or data streams is crucial for intelligent behavior and decision-making. As a fundamental quantity for measuring statistical relationships, mutual information has been widely studied and used for its generality and equitability. However, existing methods either *lack the efficiency* required for real-time applications or the *differentiability* necessary with end-to-end learning frameworks. In this paper, we present *InfoNet*, a *feed-forward* neural estimator for mutual information that leverages the attention mechanism and the computational efficiency of deep learning infrastructures. By training InfoNet to maximize a dual formulation of mutual information via a feed-forward prediction, our approach *circumvents* the time-consuming test-time optimization and comes with the capability to avoid local minima in gradient descent. We evaluate the effectiveness of our proposed scheme on various families of distributions and check its generalization to another important correlation metric, i.e., the Hirschfeld-Gebelein-Rényi (HGR) Maximal Correlation. Our results demonstrate a graceful *efficiency-accuracy* trade-off and *order-preserving* properties of InfoNet, providing a *comprehensive toolbox* for estimating both the Shannon Mutual Information and the HGR Correlation Coefficient. We will make the code and trained models publicly available and hope it can facilitate studies in different fields that require real-time mutual correlation estimation.

## 1 INTRODUCTION

We live in a universe where different entities are interconnected. For example, particles at the micro level can exhibit entanglement, which is described by quantum mechanics, and celestial bodies at the macro level are governed by gravity, which is characterized by general relativity. The presence of interconnections guarantees that our observations of the states of diverse entities around us are intricately correlated instead of independently distributed, which in turn allows us to make reasoning and predictions.

Consequently, being able to efficiently estimate the correlations between scene entities from sensory signals of the environment serves as a foundation for the emergence of intelligent behavior. *Especially*, considering an embodied agent that interacts with the scene and receives large volumes of streaming data, e.g., video, audio, and touch, within seconds. A quick estimation of correlations would help an agent build informative representations of the surroundings

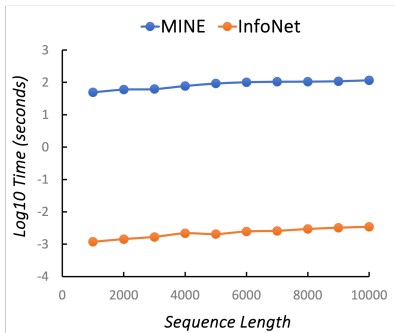

Figure 1: The log-scale run time of MINE (Belghazi et al., 2018) and the proposed InfoNet. It shows that InfoNet is constantly faster by magnitudes than MINE on sequences with different lengths.

and determine what is important for its survival. *Besides* data from embodied agents dwelling in the physical environment, each second, we generate gazillions of data across the internet, for example, the price of the stocks, messages on social media, transactions on E-commerce websites, and data from Internet-of-Things devices. Being able to perform an efficient estimation of the mutual correlations between different type or parts of the data also inform various analysis that is critical for decision-making. Thus, a *general-purpose* correlation measure between two random variables is needed given the anisotropic information contained in the data and their complex dependence.

In this paper, we study how to *neuralize* the computation of mutual information between two random variables from sequences sampled from their *empirical* joint distribution. *Specifically*, we want to explore whether the estimation of mutual information (MI) can be performed by a *feed-forward prediction* of a neural network, i.e., taking a pair of sequences as input and speeding out the MI estimate without re-training, which then guarantees efficiency and makes the estimation procedure differentiable, *enabling* the integration into end-to-end training frameworks for other applications.

As a fundamental concept in information theory (Shannon, 1948), a huge amount of effort has been devoted to the estimation of mutual information (Paninski, 2003; Kraskov et al., 2004), due to its generality and equitability (Reshef et al., 2011; Kinney & Atwal, 2014). For example, many algorithms have been proposed to improve the accuracy and efficiency of mutual information estimation, which include non-parametric methods (Moon et al., 1995; Pál et al., 2010; Marx et al., 2021) and parametric methods (Hulle, 2005; Sugiyama et al., 2012; Ince et al., 2017). However, most of them do not utilize neural networks and can not benefit from advances in deep learning techniques. Recently, MINE (Belghazi et al., 2018) employs a dual formulation of the Kullback–Leibler divergence and estimates the mutual information of a pair of sequences by optimizing a neural network's parameters against the dual objective. Even though the estimation can be performed via back-propagation, the optimization process is still behind real-time (Fig. 1, where a pair of sequences is sampled from a randomly generated mixture of Gaussian). *Moreover*, each time the joint distribution changes (different sequences), a new optimization has to be performed (e.g., the network in MINE is only optimized for a pair of sequences from a single distribution), thus not efficient.

To overcome these difficulties, yet still enjoy the *efficiency* of deep networks and their *differentiability*, we propose a novel network architecture that leverages the attention mechanism (Vaswani et al., 2017) and encodes the aforementioned optimization into the network parameters. Specifically, the proposed network takes as input a sequence of observations (pairs) and outputs a tensor, which aims at maximizing the Donsker-Varadhan (Donsker & Varadhan, 1983) dual and can be converted into the mutual information estimate by a quick summation over different entries. This way, we *transform* the optimization-based estimation into a feed-forward prediction, thus *bypassing* the time-consuming test-time gradient computation and avoiding sub-optimality via large-scale training on a *wide spectrum* of distributions. *Furthermore*, we evaluate the effectiveness of the proposed feed-forward scheme on the Hirschfeld-Gebelein-Rényi (HGR) Maximum Correlation Coefficient (Gebelein, 1941), which satisfies the seven postulates of a dependence measure Rényi (1959) and is well accepted as a correlation quantification in machine learning (Lopez-Paz et al., 2013).

In summary, we 1) propose a neural network and training method that can estimate the mutual information between two random variables (sequences) in an efficient feed-forward pass; 2) perform an extensive study of the effectiveness of the proposed scheme with different families of distribution and verify its accuracy and order-preserving properties; 3) provide a comprehensive toolbox for estimating both the Shannon Mutual Information and the HGR Maximal Correlation Coefficient with up-to-scale guarantees; and 4) validate the generalization of the proposed InfoNets on real-world distributions and show promising results in object discovery from videos.

## 2 METHOD

We aim to provide neural tools that can estimate complex nonlinear correlations between two random variables (RVs) in real-time. Particularly, we are interested in two correlation measures: Shannon's Mutual Information (MI) and the HGR Maximal Correlation Coefficient (MCC). Both are capable of measuring complicated statistical dependencies. We denote them as $\mathcal{C}^{\mathrm{info}}$ (MI) and $\mathcal{C}^{\mathrm{max}}$ (MCC), respectively, and detail their neural computation framework in the following.

**Problem Statement**    We consider the real-world scenarios where an agent keeps receiving sensory inputs via multiple channels, e.g., the motion of landmarks on an object, the temperature of the air, and the singing of a chorus. We consider these observations as random variables of any kind, and treat their (synchronized) temporal sequences as if sampled from an empirical joint distribution. More explicitly, we characterize observations $\{(x_t, y_t)\}_{t=1}^{T}$ as samples from a joint distribution $\mathrm{p}(\mathbf{x}, \mathbf{y})$, e.g., by histogramming. Our goal is to compute the correlations mentioned above between $\mathbf{x}$ and $\mathbf{y}$, i.e., either $\mathcal{C}^{\mathrm{info}}(\mathbf{x}, \mathbf{y})$ or $\mathcal{C}^{\mathrm{max}}(\mathbf{x}, \mathbf{y})$, in an efficient manner such that an agent can leverage these correlations to learn useful representations of the scene, for example, a knob controls the status of a bulb, so to make effective decisions. Specifically, we aim to train neural networks $\phi$ such that

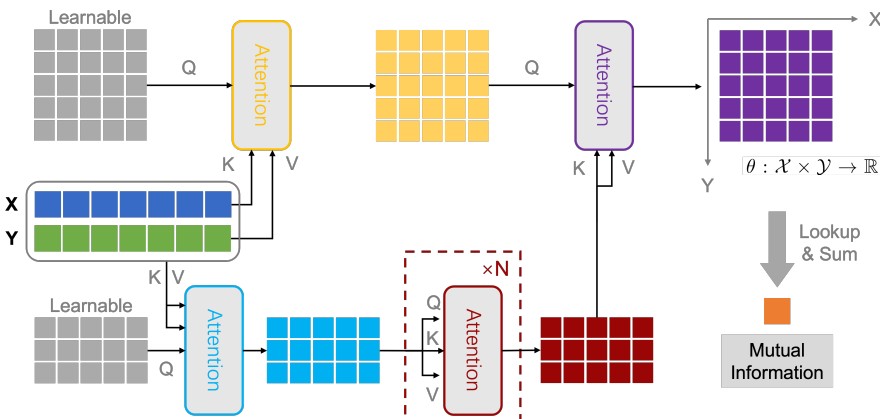

Figure 2: The proposed InfoNet architecture for the feed-forward prediction of mutual information, which consists of learnable queries and attention blocks. It takes in a sequence of samples of two random variables, and outputs a look-up table (top-right) representing a discretization of the optimal scalar function defined on the joint domain in the Donsker-Varadhan representation (Donsker & Varadhan, 1983). The mutual information between the two random variables (sequences) can then be computed by summation according to Eq. 1.

$\mathcal{C}(\mathbf{x}, \mathbf{y}) = \phi(\{(x_t, y_t)\})$ is a fast feed-forward prediction from the input sequences. In this work, we focus on the efficient computation of low-dimensional random variables, e.g., 1D/2D, and leverage the projection technique in Goldfeld & Greenewald (2021) for the extension to high-dimensional without sacrificing the efficiency.

## 2.1 FEED-FORWARD NEURAL ESTIMATION OF MUTUAL INFORMATION

Mutual information captures how much is known about one random variable by observing the other. It is usually written as the reduction of the Shannon Entropy: $\mathcal{C}^{\mathrm{info}}(\mathbf{x}, \mathbf{y}) = \mathbb{H}(\mathbf{x}) - \mathbb{H}(\mathbf{x}|\mathbf{y})$ or in the form of the Kullback–Leibler divergence (Kullback & Leibler, 1951): $\mathcal{C}^{\mathrm{info}}(\mathbf{x}, \mathbf{y}) = D_{\mathrm{KL}}(\mathrm{p}_{\mathbf{x}, \mathbf{y}} \| \mathrm{p}_{\mathbf{x}} \cdot \mathrm{p}_{\mathbf{y}})$. However, an exact computation is only tractable for discrete variables or a limited family of distributions (Paninski, 2003). Recently, MINE (Belghazi et al., 2018) proposes estimating mutual information by optimizing a neural network with gradient ascend according to a dual formula of the KL-divergence (Donsker & Varadhan, 1983). It can deal with continuous random variables, but has to train from scratch for different joint distributions $\mathrm{p}(\mathbf{x}, \mathbf{y})$, and is hardly real-time.

Here, we improve the efficiency of the dual estimation of mutual information to enable learning from a vast amount of correlations embedded in unlimited data streams. Our idea is to leverage the dual formulation yet encode the optimization as a feed-forward neural network prediction. In this way, we can speed up the estimation by magnitudes and enjoy the benefit of the differentiability of the neural networks. Next, we detail the dual formulation we employ for MI estimation and elaborate on the proposed methods for training the neural network $\phi$ for computing mutual information (correlation) between two jointly sampled sequences.

**Dual Estimation of Mutual Information** According to the Donsker-Varadhan representation (Donsker & Varadhan, 1983) (similar to the objective of contrastive learning Gutmann & Hyvärinen (2010)), we can write the KL-divergence between two distributions as: $D_{\mathrm{KL}}(\mathrm{p} \| \mathrm{q}) = \sup_\theta \mathbb{E}_{\mathrm{p}}[\theta] - \log(\mathbb{E}_{\mathrm{q}}[\exp(\theta)])$, where $\theta$ is a scalar function defined on the joint domain with finite expectations. We employ this representation and write the dual estimation formula for MI as:

$$\mathcal{C}^{\mathrm{info}}(\mathbf{x}, \mathbf{y}) = \sup_\theta \mathcal{J}^{\mathrm{info}}(\theta; \mathbf{x}, \mathbf{y}) = \sup_\theta \mathbb{E}_{\mathrm{p}_{\mathbf{x}, \mathbf{y}}}[\theta] - \log(\mathbb{E}_{\mathrm{p}_{\mathbf{x}} \cdot \mathrm{p}_{\mathbf{y}}}[\exp(\theta)]), \tag{1}$$

with $\theta : \mathcal{X} \times \mathcal{Y} \to \mathbb{R}$ and $\mathcal{X}, \mathcal{Y}$ the domain of the random variables $\mathbf{x}, \mathbf{y}$. One can instantiate $\theta$ as a neural network or a set of tunable parameters and optimize for the upper bound of the right-hand side quantity in Eq. 1 via backpropagation (Belghazi et al., 2018). The optimal value can then serve as the estimate of the mutual information between $\mathbf{x}$ and $\mathbf{y}$. To avoid costly computation, we propose to train a neural network $\phi$ that can perform the optimization in a feed-forward manner, which fully utilizes the efficiency of parallel computing units and enables efficient prediction of the supremum. In other words, we treat the scalar-valued function $\theta$ as the output of the neural network $\phi$.

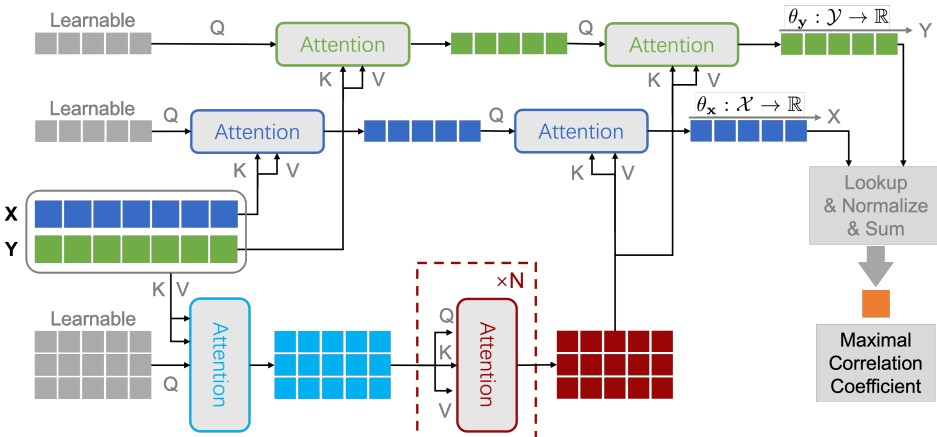

Figure 3: The proposed InfoNet architecture for the feed-forward prediction of the HGR Maximal Correlation Coefficient (MCC). The overall structure is similar to its counterpart for mutual information estimation despite that the queries to generate the lookup tables for $\theta_{\mathbf{x}}$ and $\theta_{\mathbf{y}}$ are separated. The MCC between the two random variables can then be computed by normalization (to satisfy the MCC definition) and summation according to Eq. 3.

**Feed-Forward MI Estimator**    To enable feed-forward prediction, we let $\theta_{\mathbf{x},\mathbf{y}} = \phi(\{(x_t, y_t)\}) \in \mathbb{R}^{L \times L}$ be a 2D tensor, where $L$ represents the quantization levels of the range of the involved random variables. Now, the value of $\theta_{\mathbf{x},\mathbf{y}}(x_t, y_t)$ can be directly read out from the tensor as a look-up table with correct indexing and appropriate interpolation.

To facilitate the computation, we design a neural network by adapting the attention mechanism described in (Jaegle et al., 2021). We illustrate the proposed feed-forward structure $\phi$ in Fig. 2. It takes in a pair of joint sampled sequences, e.g., $\{(x_t, y_t)\}_{t=1}^{T}$, and outputs a tensor $\theta_{\mathbf{x},\mathbf{y}}$ as a discretization of the scalar function $\theta$ in Eq. 1. To account for the fact that the value of a random variable could be arbitrary, we normalize the sequences (before giving them to the network) to be within the range of $[-1, 1]$, so the quantization granularity would be $2/L$. Please note that if the same scaling factor is applied to both random variables, the mutual information between them will not change. This is because MI measures the amount of information one variable contains about the other, which is invariant under bijection. More explicitly, the effect on the entropies caused by the scaling factor is canceled out as mutual information is the difference between two entropies (one is conditional). With the predicted (discretized) function $\theta_{\mathbf{x},\mathbf{y}}$, we can then compute an estimate of the mutual information between $\mathbf{x}$ and $\mathbf{y}$ using the quantity $\mathcal{J}^{\text{info}}(\theta; \mathbf{x}, \mathbf{y})$ in Eq. 1.

However, the prediction is only accurate when the lookup table $\theta_{\mathbf{x},\mathbf{y}}$ maximizes the right-hand side of Eq. 1. To ensure the estimation quality, we train the neural network $\phi$ using the following training objective:

$$\mathcal{L}^{\text{info}}(\phi, \mathcal{D}) = \frac{1}{N} \sum_{i=1}^{N} \mathcal{J}^{\text{info}}(\theta_{\mathbf{x}^i, \mathbf{y}^i}; \mathbf{x}^i, \mathbf{y}^i)$$
$$= \frac{1}{N} \sum_{i=1}^{N} \left\{ \frac{1}{T} \sum_{t=1}^{T} \theta_{\mathbf{x}^i, \mathbf{y}^i}(x_t^i, y_t^i) - \log\left( \frac{1}{T} \sum_{t=1}^{T} \exp(\theta_{\mathbf{x}^i, \mathbf{y}^i}(x_t^i, \tilde{y}_t^i)) \right) \right\}. \quad (2)$$

Here $\mathcal{D}$ is a dataset of N different distributions, i.e., $\mathcal{D} = \{(\mathbf{x}^i, \mathbf{y}^i)\}_{i=1}^{N}$ with each $(\mathbf{x}^i, \mathbf{y}^i) = \{(x_t^i, y_t^i)\}_{t=1}^{T}$ represents an empirical distribution $\mathrm{p}(\mathbf{x}^i, \mathbf{y}^i)$. And $\tilde{y}_i$ can be sampled from the marginal distribution $\mathrm{p}(\mathbf{y}^i)$ by simply breaking the pairing. We detail the generation of the training data in Sec. 3.1. Note that the training is performed with all possible (joint) empirical distributions between $\mathbf{x}$ and $\mathbf{y}$ in contrast to a single one in Eq. 1, and $\theta_{\mathbf{x}^i, \mathbf{y}^i}$ is supposed to maximize the quantity $\mathcal{J}^{\text{info}}$ for every $\mathrm{p}(\mathbf{x}^i, \mathbf{y}^i)$. Thus, the network $\phi$ has to learn the optimization of $\mathcal{J}^{\text{info}}$ via its parameters. Please refer to Sec. 3 for more training details.

## 2.2 Feed-Forward Neural Estimation of Maximal Correlation Coefficient

It is widely accepted that mutual information characterizes correlations without bias for different dependency types (Kinney & Atwal, 2014). However, the HGR Maximal Correlation Coefficient (MCC) (Gebelein, 1941) is shown to satisfy the seven postulates that a dependence measure should have (Rényi, 1959; Bell, 1962). For example, one of the postulates is that the dependence measure should be in the range of $[0, 1]$. This normalization characteristic is useful when comparing and interpreting the dependencies between different pairs of random variables, and it is not possessed by mutual information since its range varies with the entropy of the random variables. Moreover, a zero maximal correlation coefficient serves as a necessary and sufficient condition for statistical independence between two random variables.

To provide a comprehensive toolbox for efficient estimation of mutual correlation between data streams, we also explore the possibility of neuralizing the computation of the MCC between two random variables $\mathbf{x}, \mathbf{y}$. Given $\theta_\mathbf{x} : \mathcal{X} \to \mathbb{R}$ and $\theta_\mathbf{y} : \mathcal{Y} \to \mathbb{R}$, The MCC is defined as:

$$\mathcal{C}^{\max}(\mathbf{x}, \mathbf{y}) = \sup_{\theta_\mathbf{x}, \theta_\mathbf{y}} \mathcal{J}^{\max}(\theta_\mathbf{x}, \theta_\mathbf{y}; \mathbf{x}, \mathbf{y}) = \sup_{\theta_\mathbf{x}, \theta_\mathbf{y}} \mathbb{E}_{p_{\mathbf{x}, \mathbf{y}}}[\theta_\mathbf{x} \theta_\mathbf{y}],$$
$$\text{s.t. } \mathbb{E}_{p_\mathbf{x}}[\theta_\mathbf{x}] = \mathbb{E}_{p_\mathbf{y}}[\theta_\mathbf{y}] = 0 \text{ and } \mathbb{E}_{p_\mathbf{x}}[\theta_\mathbf{x}^2] = \mathbb{E}_{p_\mathbf{y}}[\theta_\mathbf{y}^2] = 1. \quad (3)$$

Similar to the Donsker-Varadhan representation of mutual information, MCC estimation is also carried out as an optimization. However, MCC comes with the constraints that the mean and variance of $\theta$'s should be zero and unit, which guarantees the normalization postulate by Rényi (1959). To leverage the feed-forward efficiency of neural networks, rather than solving an optimization whenever the RVs switch, we apply the same design principles in MI estimation for computing MCCs. Again, we denote $\phi$ as the neural network that is trained to perform the optimization and output the optimizers for $\mathcal{J}^{\max}$ conditioned on the input observations $\{(x_t^i, y_t^i)\}$.

**Feed-Forward MCC Estimator** Following the MI prediction scheme, we discretize the range of the involved random variables into $L$ levels. Correspondingly, $[\theta_\mathbf{x}, \theta_\mathbf{y}] = \phi(\{(x_t, y_t)\})$ are the look-up tables with $\theta_\mathbf{x}, \theta_\mathbf{y} \in \mathbb{R}^L$. The network architecture of $\phi$ for MCC computation is shown in Fig. 3. In contrast to its counterpart for mutual information estimation, the queries for generating the lookup tables are computed with two separate branches consisting of attention blocks for efficiency. Note that since the two lookup tables are defined on a single domain instead of the joint domain of $\mathbf{x}$ and $\mathbf{y}$, we can increase the quantization levels without incurring much computational overhead. This characteristic of computing MCCs allows us to tune $L$ in a much wider range for studying the effect of the quantization.

To make sure that the feed-forward prediction outputs lookup tables that deliver an accurate estimate of the MCC, we train the neural network $\phi$ using the following objective according to Eq 3:

$$\mathcal{L}^{\max}(\phi, \mathcal{D}) = \frac{1}{N} \sum_{i=1}^{N} \mathcal{J}^{\max}(\theta_{\mathbf{x}^i}, \theta_{\mathbf{y}^i}; \mathbf{x}^i, \mathbf{y}^i) = \frac{1}{N \times T} \sum_{i=1}^{N} \sum_{t=1}^{T} \mathcal{M}(\theta_{\mathbf{x}^i}(x_t^i)) \mathcal{M}(\theta_{\mathbf{y}^i}(y_t^i)), \quad (4)$$

where $\mathcal{M}$ represents the normalization operator that ensures compatibility with the definition of HGR Maximal Correlation Coefficient in Eq. 3. It is worth noting that the normalization should be performed after the look-up operations instead of directly applied to the look-up tables themselves, which helps enforce that the mean and variance of the sequences are put to zero and one since we are dealing with empirical distributions.

## 3 Training Algorithm

Next, we detail the generation of the training data and the implementation for reproducibility.

### 3.1 Data Generation

To generate training data, we consider sampling the joint sequences $\mathcal{D} = \{(\mathbf{x}^i, \mathbf{y}^i)\}_{i=1}^{N}$ from the Gaussian Mixture Models (GMMs). It is widely accepted that GMMs are a versatile and effective tool for modeling real-world distributions due to their capability to handle complex and noisy data. Specifically, GMMs represent a family of distributions as a weighted sum of Gaussian components and are defined as: $p(z) = \sum_{i=1}^{K} \pi_i \mathcal{N}(z | \mu_i, \Sigma_i)$, where $p(z)$ is the probability density function

---

**Algorithm 1** InfoNet Training

---

**Require:** A maximum number of Gaussian components; Step size $\eta$
1: **Repeat:**
2: Randomly select $N$ two-dimensional Gaussian mixture distributions
3: Select $T$ data points from each distribution as joint distribution
4: Shuffle joint samples to get marginal samples
5: Put joint samples into the model and get $N$ two-dimension lookup tables
6: Apply lookup function to get the corresponding $\theta_{\mathbf{x}^i, \mathbf{y}^i}(x_t^i, y_t^i)$ and for all data points in joint samples and marginal samples
7: $\mathcal{L} \leftarrow \frac{1}{N} \sum_{i=1}^{N} \left\{ \frac{1}{T} \sum_{t=1}^{T} \theta_{\mathbf{x}^i, \mathbf{y}^i}(x_t^i, y_t^i) - \log\left( \frac{1}{T} \sum_{t=1}^{T} \exp(\theta_{\mathbf{x}^i, \mathbf{y}^i}(x_t^i, \tilde{y}_t^i)) \right) \right\}$.
8: Do gradient ascent for $\mathcal{L}$
9: **Until Convergence.**

---

(PDF) of the GMM, $K$ is the total number of components in the mixture, $\pi_i$ denotes the weight of the $i$-th component satisfying $\sum_{i=1}^{K} \pi_i = 1$, and $\mathcal{N}(z|\mu_i, \Sigma_i)$ is the PDF of a Gaussian with mean $\mu_i$ and covariance $\Sigma_i$. By fitting the parameters $K$, $\pi_i$, $\mu_i$, and $\Sigma_i$, a GMM can faithfully approximate an arbitrary target distribution. We argue that sampling from GMMs is necessary due to two facts: 1) we can not guarantee enough coverage of real-world distributions with a limited budget; 2) we can synthesize arbitrarily complex distributions using GMMs so that the trained InfoNet can generalize to real-world ones (in a similar spirit to Cranmer et al. (2020); Lavin et al. (2021)).

We set the maximum number of components to 20 to ensure enough diversity in the sampled GMMs. Specifically, we first randomly choose a number $K$ from $\{1, 2, ..., 20\}$, and then we perform another sampling of the component weights $\{\pi_i\}_{i=1}^{K}$ such that their sum is one. For each GMM component, we randomly sample its mean from the interval $[-5, 5]$. To generate the covariance matrix, we begin by creating a matrix $\mathbf{D}$ where each element is sampled from the range $[-3, 3]$. Then, the covariance matrix is derived by $\Sigma = \mathbf{D}\mathbf{D}^T + \epsilon \mathbf{I}$, where $\epsilon = 0.01$ is utilized to enforce the matrix to be positive definite. To this end, a random GMM distribution is instantiated, and we can sample from it to get two sequences by partitioning $z$ into two parts. The jointly sampled GMM sequences can be found in Sec. A. Finally, in each training batch, we have 32 randomly generated GMM sequences, each having a length equal to 2000. Please note that, for each batch, we sample a separate set of GMMs to make sure the training data for InfoNet is diverse and the training can explore the whole space of the GMM family. Trained with randomly sampled distributions, our model should be capable of estimating mutual information for real-world distributions encountered during inference.

### 3.2 NETWORK TRAINING

The network architectures are illustrated in Fig. 2 (InfoNet-MI) and Fig. 3 (InfoNet-MCC). To train, we normalize the sample values to $[-1, 1]$. This ensures that the MI or MCC between the random variables is unchanged. Also, normalizing different sequences into the same range allows more efficient training and gives an easy interface to real-world data, whose range can be arbitrary. Additionally, we apply a bilinear interpolation to get the values of $\theta_{\mathbf{x}, \mathbf{y}}$ on non-grid points. Please check Algorithm 1 for the detailed training procedure.

## 4 EXPERIMENTS

We focus on studying three different aspects related to the effectiveness of the training and the efficiency of the estimation: 1) the evaluation criteria and collection of evaluation data for the proposed InfoNets and baseline methods; 2) the comparison with other baseline methods in various settings and validation of the effectiveness of the proposed feed-forward estimation of mutual correlations; and 3) conducting experiments on real-world data to assess performance against other baseline methods in terms of efficiency and generalization.

### 4.1 EVALUATION DATA AND METRICS

**Evaluation** The evaluation sequences are generated in the same manner as the training ones. To get the ground-truth mutual information for the evaluation sequences, we consider two situations. If the sequences come from a single-component GMM, then we apply the analytical formula of MI for

Gaussian. Otherwise, we apply the Monte-Carlo Integrate (MCI) method (Shapiro, 2003) to compute the ground truth from the samples. The MCI method first estimates the GMM parameters from the sequences and then leverages a numerical tool to compute the MI between the random variables represented by the estimated GMM. Note that for HGR-MCC, we do not have an integration formula as MI, so the ground truth is not explicitly defined, whose evaluation is detailed in the following.

**Setup and Metrics**   We evaluate our method and others with the following setups.

- **Sanity Check**. We use the sequences sampled from single-component Gaussian distributions to benchmark different methods, which is a common evaluation scenario adopted by most MI estimation methods. The estimated MI values are directly compared to the GT computed using the analytical formula.

- **On GMMs with Multiple Components**. To account for the noise in the estimated GT for GMMs with multiple components, we study the mean and variance of the errors of the estimated MI. Specifically, we divide the sampled evaluation sequences into several categories according to their ground-truth MI values, e.g., sequences with ground-truth MI around 0.5. We report the mean and variance of the errors for different methods.

- **Mutual Correlation Order Accuracy**. Beyond application domains where the exact MI value is critical, most of the time, for decision-making, the more important is the order of mutual correlations between different random variables. For this, we generate an evaluation dataset consisting of triplets of random variables $\{(\mathbf{x}, \mathbf{y}, \mathbf{y}')\}$, whose ground truth order is determined by the computed GT mutual information (i.e., $\mathbb{I}(\mathbf{x}, \mathbf{y}) > \mathbb{I}(\mathbf{x}, \mathbf{y}')$). We apply different methods on the triplets to test the correlation order accuracy averaged over all triplets.

- **Generalization to High-Dimensional and Real-World Distributions.** We verify the generalization of the trained InfoNets on real-world data (e.g., Radford et al. (2021); Zheng et al. (2023)), where the goal is to check whether the points coming from the same object in motion can be grouped correctly by looking at the estimated mutual correlation, as well as how InfoNet works with high-dimensional out-of-domain data leveraging Goldfeld & Greenewald (2021).

## 4.2   RESULTS AND COMPARISON

In this section, we report the experimental results and comparisons between the proposed InfoNet for feed-forward mutual correlation estimation and other methods. We consider three major baselines: KSG (Kraskov et al., 2004), which computes MI based on entropy estimates by averaging k-nearest neighbor distances of the data points; KDE (Silverman, 2018), which estimates the joint and marginal probability density with kernel functions and then computes the MI by integration; and MINE (Belghazi et al., 2018), which estimates the MI with the same dual formulation as ours but resorts to optimizing a network for different distributions in contrast to the feed-forward prediction of InfoNet. All the evaluations are conducted on an RTX 4090 GPU with an AMD Ryzen Threadripper PRO 5975WX 32-Core CPU.

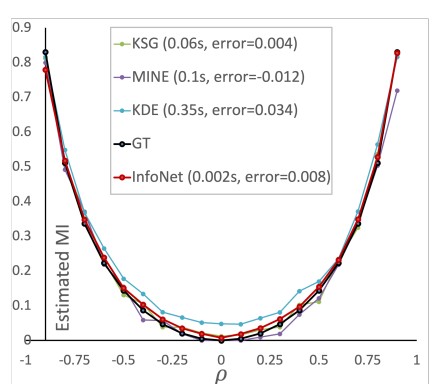

Figure 4: Comparison of MI estimates with Gaussian (runtime included).

**Sanity Check on Gaussian**   We perform a check on the Gaussian distributions. In this case, the mutual information between random variables strongly depends on their Pearson correlation coefficient $\rho$. The evaluation with this setup is commonly reported in other studies, allowing us to verify if the trained InfoNet is working properly.

For fair comparisons, we train the MINE model with a batch size equal to 500 and for a total of 500 steps with learning rate 0.001. For the KSG method, we choose a neighborhood size of $k = 5$ for optimal performance. As depicted in Fig. 4, the trained InfoNet can predict the ground-truth mutual information more faithfully than the other baselines. The mean error for each method can be found in the legend inserted in the figure. We can see that InfoNet quantitatively achieves a similar error with KSG but is thirty times faster. When compared to MINE, InfoNet runs 50 times faster, while

Table 1: Error mean and variance of different MI estimators. Methods that do not rely on neural networks are highlighted in Blue, and those leveraging neural networks are colored Green.

|  | MI | 0.0 | 0.1 | 0.2 | 0.3 | 0.4 | 0.5 | 0.6 | 0.7 | 0.8 | 0.9 |
|---|---|---|---|---|---|---|---|---|---|---|---|
| **Mean** | KSG | 0.001 | 0.001 | 0.004 | 0.006 | 0.008 | 0.009 | 0.012 | 0.015 | 0.016 | 0.014 |
|  | KDE | 0.005 | 0.010 | -0.003 | -0.350 | -0.071 | -0.109 | -0.155 | -0.199 | -0.239 | -0.292 |
|  | MINE-500 | **-0.003** | **-0.058** | -0.116 | -0.173 | -0.228 | -0.294 | -0.344 | -0.399 | -0.431 | -0.485 |
|  | MINE-100 | -0.008 | -0.092 | -0.173 | -0.251 | -0.336 | -0.420 | -0.504 | -0.584 | -0.658 | -0.742 |
|  | InfoNet | 0.018 | 0.010 | **0.0001** | **-0.026** | **-0.056** | **-0.087** | **-0.125** | **-0.155** | **-0.193** | **-0.233** |
| **Variance** | KSG | 2e-4 | 3e-4 | 4e-4 | 5e-4 | 6e-4 | 8e-4 | 9e-4 | 9e-4 | 1e-3 | 1e-3 |
|  | KDE | 0.010 | 0.005 | 0.001 | 0.003 | 0.004 | 0.005 | 0.010 | 0.012 | 0.014 | 0.019 |
|  | MINE-500 | **4e-5** | 0.001 | 0.004 | 0.008 | 0.013 | 0.018 | 0.027 | 0.039 | 0.052 | 0.060 |
|  | MINE-100 | 4e-5 | **5e-4** | 0.002 | 0.005 | 0.009 | 0.012 | 0.017 | 0.025 | 0.033 | 0.040 |
|  | InfoNet | 4e-4 | 0.001 | **0.001** | **0.002** | **0.004** | **0.006** | **0.010** | **0.014** | **0.020** | **0.028** |

achieving a 30% percent improvement in accuracy. This sanity check verifies that the proposed InfoNet has an optimal efficiency-accuracy tradeoff than others.

**On GMMs with Multiple Components** We perform evaluations on GMMs with multiple components, which is a more challenging but practical task. We generate a test dataset using the following methodology: Firstly, we establish 10 levels for mutual information, ranging from 0.0 to 0.9. Next, we employ the approach mentioned in training data generation to create random GMM distributions. Meanwhile, the mutual information between the two random variables is calculated using the MCI method. If the computed mutual information falls within a narrow range of $\pm 0.02$ around any of the 10 representative level values, we assign the corresponding GMM distribution to that level and record the precise mutual information value at the same time. The process proceeds until each level of mutual information is assigned 1000 distributions. Subsequently, we sample sequences of length 2000 for each recorded GMM distribution, with each containing values for both random variables. We then estimate the MI of each sequence using different methods. The error mean and variance for each method on different MI levels are summarized in Tab. 6.

Accordingly, we can make the following observations: 1) Even though traditional methods (blue) can not utilize neural networks for computational efficiency, they perform relatively well in terms of mean error and variance (KSG); 2) KDE also has small variances but has larger mean errors than InfoNet. 3) Among the neural methods (InfoNet and variants of MINE), our model achieves much smaller mean errors, and the prediction is more stable than MINE, which is tested with both 500 and 100 steps during the test-time training. The runtime for MINE-100 and MINE-500 are 0.17 and 0.991 seconds, respectively, while the runtime for InfoNet is 0.008 seconds.

**Mutual Correlation Order Accuracy** Now we report the performance of different methods measured by the correlation order accuracy. We consider the study under different numbers of GMM components, i.e., $K$ ranges from 1 to 10, so we can get an idea of how the accuracy varies as the difficulty of estimating the mutual correlation increases. We collect 2000 triplets mentioned above (Sec. 4.1) for each of the different categories. For each triplet, if the estimated order (either $\mathbb{I}(\mathbf{x}, \mathbf{y}) > \mathbb{I}(\mathbf{x}, \mathbf{y}')$ or $\mathbb{I}(\mathbf{x}, \mathbf{y}) \leq \mathbb{I}(\mathbf{x}, \mathbf{y}')$) matches with the ground truth computed by the MCI method, it is considered as accurately ordered. We report the results in Tab. 7. We can see that the order accuracy of InfoNet is unanimously higher than the test-time optimization method (MINE) even though both employ neural networks. Also, as the estimation difficulty increases, InfoNet still outputs accurate estimates of the orders between different random variables measured by the ground truth mutual information, justifying our model as a reliable correlation estimator for making decisions based on the correlation order.

**High-Dimensional and Real-World Data** Due to limited space, please refer to Sec. A.3 for the clustering results on motion data with the proposed correlation. For high-dimensional MI estimation results, please refer to Sec. B.3.1 and Sec. B.3.2. Please also refer to Sec. B.3.3 for results on high-dimensional and real-world data (images). Generalization on more distributions other than GMMs is presented in Sec. B.2.

## 5 RELATED WORK

Mutual information estimation measures the statistical dependence between variables using various nonparametric and parametric approaches. Nonparametric methods, such as K-Nearest Neighbors

Table 2: Correlation order accuracy of different MI estimators. Methods that do not rely on neural networks are highlighted in Blue, and those leveraging neural networks are colored Green.

| NO. OF COMPS. | 1 | 2 | 3 | 4 | 5 | 6 | 7 | 8 | 9 | 10 |
|---|---|---|---|---|---|---|---|---|---|---|
| KSG | 98.7 | 99.0 | 98.2 | 98.0 | 97.9 | 97.7 | 97.6 | 97.5 | 97.0 | 97.3 |
| KDE | 97.4 | 97.7 | 97.9 | 97.5 | 97.9 | 97.8 | 97.0 | 97.4 | 97.4 | 97.4 |
| MINE-500 | **98.5** | 91.2 | 90.8 | 87.2 | 84.5 | 83.7 | 81.2 | 79.6 | 81.3 | 78.1 |
| MINE-100 | 94.6 | 77.1 | 75.4 | 71.6 | 67.5 | 69.4 | 66.5 | 66.3 | 68.7 | 66.4 |
| MINE-10 | 60.9 | 56.1 | 55.1 | 54.3 | 52.4 | 54.9 | 53.7 | 50.4 | 53.1 | 52.5 |
| INFONET | 97.3 | **96.2** | **97.0** | **97.5** | **97.1** | **97.6** | **97.2** | **97.2** | **97.8** | **97.4** |

(KNN) and Kernel Density Estimation (KDE), estimate mutual information without assuming specific probability distributions (Reshef et al., 2011; Kinney & Atwal, 2014; Khan et al., 2007; Kwak & Choi, 2002; Kraskov et al., 2004; Pál et al., 2010; Gao et al., 2015b; 2017; Runge, 2018; Lord et al., 2018; Moon et al., 1995; Steuer et al., 2002; Gretton et al., 2005; Kumar et al., 2021). However, these methods have limitations such as sensitivity to parameter choice, curse of dimensionality, computational complexity, and assumptions about continuity (Suzuki et al., 2008; Walters-Williams & Li, 2009; Gao et al., 2018; Mukherjee et al., 2020; Fukumizu et al., 2007; Estévez et al., 2009; Bach, 2022). Binning methods and adaptive partitioning are nonparametric alternatives but suffer from limitations in bin/partition selection and curse of dimensionality (Lugosi & Nobel, 1996; Darbellay & Vajda, 1999; Cellucci et al., 2005; Fernando et al., 2009; Cakir et al., 2019; Marx et al., 2021; Thévenaz & Unser, 2000; Paninski, 2003; Knops et al., 2006; Tsimpiris et al., 2012). Parametric methods assume specific distributions, e.g., Gaussian, but their accuracy relies on correct assumptions and parameter estimation (Hulle, 2005; Gupta & Srivastava, 2010; Sugiyama et al., 2012; Gao et al., 2015a; Ince et al., 2017; Suzuki et al., 2008; Walters-Williams & Li, 2009).

When dealing with limited sample size, measuring and optimizing mutual information can be challenging (Treves & Panzeri, 1995; Bach & Jordan, 2002; McAllester & Stratos, 2020). However, alternative measurements within a Reproducing Kernel Hilbert Space (RKHS) have shown effectiveness in detecting statistical dependence (Gretton et al., 2005). A kernel method using the HGR maximal correlation coefficient to capture higher-order relationships is in (Bach & Jordan, 2002). While traditional Pearson correlation focuses on linearity, HGR maximal correlation captures nonlinear dependencies. The Sigular Value Decomposition (SVD) (Anantharam et al., 2013; Makur et al., 2015), Alternating Conditional Expectation (ACE) algorithm Breiman & Friedman (1985); Buja (1990); Huang & Xu (2020); Almaraz-Damian et al. (2020), and rank correlation (Kendall, 1938; Klaassen & Wellner, 1997) are traditional methods that commonly used. Recently, neural network methods are also proposed (Xu & Huang, 2020). Maximal correlation coefficient estimation has limitations compared to mutual information. It is sensitive to linearity, limited to bivariate cases, and computationally complex. Mutual information, on the other hand, measures overall dependence, extends to multivariate settings, and is computationally efficient for discrete variables.

While there are many works on the scalable computation of MI and statistical dependences Lopez-Paz et al. (2013); Mary et al. (2019); Goldfeld & Greenewald (2021); Chen et al. (2022), our InfoNet provides an orthogonal alternative. Instead of pursuing a more accurate approximation of the highly nonlinear MI or devising more advanced yet computationally friendly correlation metrics, InfoNet focuses on MI/MCC estimation by encoding the optimization of their dual objectives into neural networks through pertaining, which allows test-time feed-forward prediction and conceptually enables more efficient and accurate solutions to these complex correlation measures. The proposed is also related to simulation-based intelligence Cranmer et al. (2020); Ramon et al. (2021). Due to limited space, please refer to Sec. B.4 for elaborations.

## 6 DISCUSSION

We propose a method that can efficiently estimate the mutual information between two random variables by looking at their jointly sampled sequences. Our method is magnitude faster than the neural counterparts that still solve an optimization during test-time. Our model is trained to estimate the mutual correlations in a feed-forward manner and can be run in real time. We verified its efficiency and effectiveness with a wide range of distribution families as well as its generalization on real-world data. We expect the proposed method and trained models can facilitate applications that require estimating a vast amount of correlation in a very low time budget.

**Ethics Statement:** The data used in our experiments are either synthesized or from public benchmarks, and all comply with the ethical regulations.

**Reproducibility Statement:** We have provided details on the network structure and the data generation procedures. We assure the readers that the training and evaluation are fully reproducible and we promise to release the repository upon completion of the review process.

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

# A    APPENDIX

## A.1    RUNTIME COMPARISON ON GMM DATA

In this section, we conduct a comparison of the time complexity between our InfoNet model and other baseline methods across different numbers of samples.

Table 3: Comparison on time complexity on Gaussian mixture distributions (Unit: seconds)

| NO. OF DATA SAMPLES | 200 | 500 | 1000 | 2000 | 5000 | 10000 |
|---|---|---|---|---|---|---|
| KSG(K=1) | 0.009 | 0.024 | 0.049 | 0.098 | 0.249 | 0.502 |
| KSG(K=5) | 0.010 | 0.025 | 0.049 | 0.102 | 0.253 | 0.513 |
| KDE | 0.004 | 0.021 | 0.083 | 0.32 | 1.801 | 6.72 |
| MINE(2000 ITERS) | 3.350 | 3.455 | 3.607 | 3.930 | 4.157 | 5.755 |
| MINE(500 ITERS) | 0.821 | 0.864 | 0.908 | 0.991 | 1.235 | 1.668 |
| MINE(10 ITERS) | 0.017 | 0.017 | 0.019 | 0.021 | 0.027 | 0.035 |
| OURS(BATCHSIZE 1) | 0.010 | 0.010 | 0.011 | 0.011 | 0.013 | 0.015 |
| OURS(BATCHSIZE 16) | **0.001** | **0.002** | **0.002** | **0.002** | **0.003** | **0.004** |

In Tab. 3, we present the results of the running time for different numbers of samples. The reported values are averaged over 100 experimental trials. For the MINE method, we set the batch size as 100 and the learning rate as 0.001. In our InfoNet method, the batch size refers to the number of distributions estimated simultaneously. The results indicate that our InfoNet model achieves remarkably faster processing speed compared to other methods across all tested sample sizes. This highlights the efficiency and effectiveness of our approach in handling various data sample scenarios.

## A.2    SMOOTHING THE LOOKUP TABLE

We have tried two smoothing techniques to enhance the smoothness of the lookup table and at last, choose method 1 in our final presented results.

Method 1: We apply a convolution layer with a non-learnable Gaussian kernel with size 15 and sigma 3 on the lookup-table layer. Figure 5 shows the visualization results of adding the Gaussian smooth kernel. The table is indeed much smoother than not smoothed.

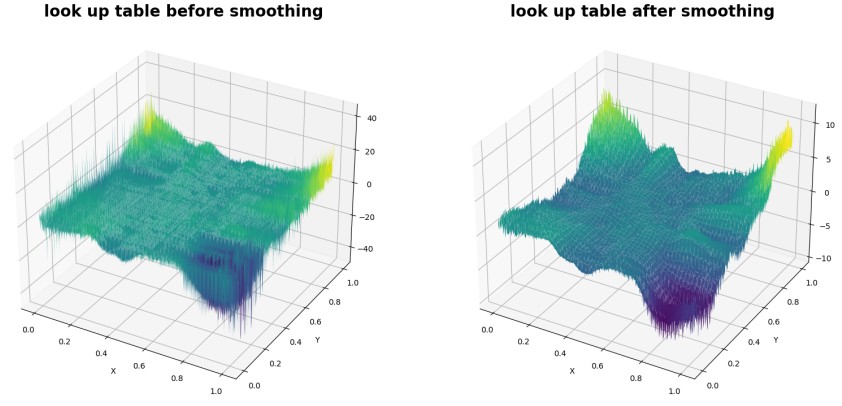

Figure 5: Comparison between the un-smoothed lookup table and smoothed lookup table.

Method 2: We add a penalty term to punish the jumps in values between adjacent points using the Laplacian operator. We add $\mathcal{L}_{smooth} = \alpha \, |\text{Laplacian(lookup-table)}|$ after Eq. 2, where Laplacian(lookup-table) can be obtained by applying a convolution layer on the lookup-table using

Laplacian kernel:

$$\text{Laplacian Kernel} = \begin{bmatrix} 0 & 1 & 0 \\ 1 & -4 & 1 \\ 0 & 1 & 0 \end{bmatrix} \tag{5}$$

### A.3 VALIDATION ON OUT-OF-DOMAIN MOTION DATA

In the field of computer vision, mutual information has greatly influenced the formulation of video object segmentation, making it an important area of study. In this section, we assess our model's performance in video segmentation tasks and present our results using Precision-Recall graphs.

Our objective is to partition the initial frame of a video into two distinct components: the foreground and the background. To achieve this, we leverage the mutual information derived from the trajectory of points within the video.

Precision-Recall graphs serve as valuable tools for evaluating the performance of segmentation models, providing key metrics for assessment. The essential definitions associated with Precision-Recall graphs are as follows:

$$\text{Precision} = \frac{\text{True Positives}}{\text{True Positives} + \text{False Positives}}, \quad \text{Recall} = \frac{\text{True Positives}}{\text{True Positives} + \text{False Negatives}} \tag{6}$$

In the segmentation task, given a mask generated by our model and the corresponding ground truth, True Positives (TP) are the number of points that exist in both the mask and the ground truth. The denominator of Precision and Recall represents the total number of points in the mask and the ground truth, respectively.

For our study, we utilize the Pointodyssey dataset (Zheng et al., 2023), which comprises a lengthy sequence of synthetic videos. This dataset offers a substantial amount of trajectory information as ground truth. However, it is worth noting that certain trajectories provided may contain unreasonable values such as "inf" or "-50000". To address this issue, we initially apply a filtering process to ensure that only points appearing throughout the entire video are considered for analysis.

We begin by selecting a point in the first frame that possesses a trajectory. Using our pre-trained MCC model, which is exclusively trained on Gaussian mixture distributions without any additional training steps, we estimate the Maximal Correlation Coefficient (MCC) between the selected point and all other points based on their trajectories. Let's denote the chosen point as $P_{idx}$. By utilizing the ground truth dataset, we can obtain the location of $P_{idx}$ in all subsequent frames. Next, we calculate the MCC between the $x$-position of $P_{idx}$ and the $x$-position of each additional point. The same process is repeated for the $y$-positions. We then compute the average of the two MCC values, resulting in $\frac{1}{2}(MCC_x + MCC_y)$, which serves as a measure for video segmentation. In Figure 6a and Figure 6b, we present the visualization of our estimated MCC values in the first frame.

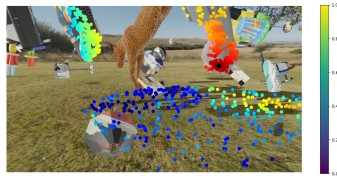 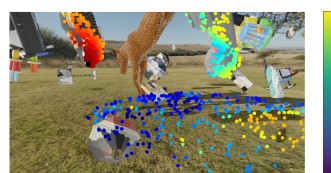

(a) Estimated MCC with point in object 1 (high-lighted black).

(b) Estimated MCC with point in object 2 (high-lighted black).

Figure 6: Visualization results using our MCC model.

Subsequently, we define a threshold parameter $\gamma$ ranging from $0$ to $1$ with an interval of $0.01$. Points with estimated MCC values greater than the threshold are considered as foreground points, while the remaining points are classified as background. By comparing this segmentation with the ground truth mask, we can compute precision and recall metrics for 7 objects from four different videos. Finally, we calculate the average precision and recall to present our overall results.

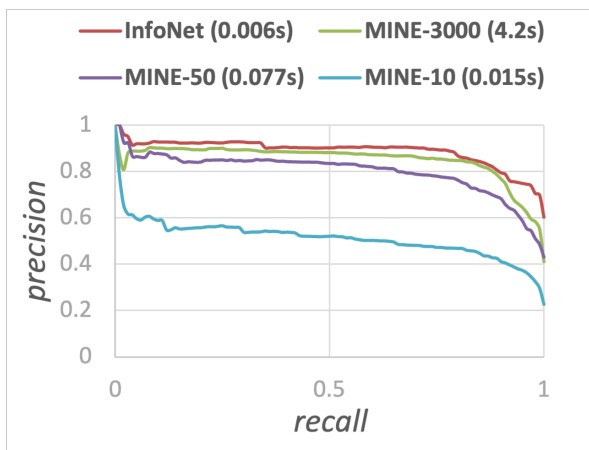

Figure 7: Averaged PR graphs of our model and MINE.

Table 4: Comparison of error mean and variance for given MI value within the range of $\pm 0.01$

| | MI | 0.0 | 0.1 | 0.2 | 0.3 | 0.4 | 0.5 | 0.6 | 0.7 | 0.8 | 0.9 |
|---|---|---|---|---|---|---|---|---|---|---|---|
| Mean | KSG | -3e-4 | 0.002 | 0.004 | 0.005 | 0.009 | 0.009 | 0.012 | 0.014 | 0.016 | 0.015 |
| | KDE | 0.007 | 0.012 | -0.0004 | -0.036 | -0.070 | -0.108 | -0.157 | -0.200 | -0.242 | -0.298 |
| | MINE/500 | -0.002 | -0.057 | -0.116 | -0.176 | -0.228 | -0.293 | -0.341 | -0.406 | -0.439 | -0.501 |
| | MINE/100 | -0.004 | -0.91 | -0.171 | -0.252 | -0.336 | -0.421 | -0.502 | -0.591 | -0.663 | -0.754 |
| | Ours | **0.016** | **0.011** | **0.0003** | **-0.025** | **-0.055** | **-0.087** | **-0.125** | **-0.157** | **-0.198** | **-0.240** |
| Variance | KSG | 2e-4 | 3e-4 | 4e-4 | 5e-4 | 6e-4 | 8e-4 | 9e-4 | 9e-4 | 1e-3 | 1e-3 |
| | KDE | 0.001 | 0.004 | 0.001 | 0.003 | 0.004 | 0.005 | 0.012 | 0.012 | 0.014 | 0.019 |
| | MINE/500 | 2e-5 | 0.001 | 0.004 | 0.008 | 0.013 | 0.018 | 0.028 | 0.042 | 0.052 | 0.057 |
| | MINE/100 | 1e-5 | 4e-4 | 0.002 | 0.005 | 0.009 | 0.012 | 0.018 | 0.024 | 0.030 | 0.037 |
| | Ours | **3e-4** | **0.001** | **0.001** | **0.002** | **0.004** | **0.006** | **0.011** | **0.013** | **0.022** | **0.031** |

In Figure 7, we present a comparison between our model and MINE using different training iterations (100, 500, 3000). The results demonstrate that our model maintains high segmentation quality while significantly reducing the required time. On average, our model takes only 6 seconds to obtain the segmentation of one video, whereas MINE requires approximately 1 minute.

### A.4 ADDITIONAL STATISTICS OF THE RESULTS

In Table 4 and Table 5, we present additional results for the error test based on the given mutual information values. However, there is a slight difference compared to the table provided in the experiment section, since we choose distributions used for the mean calculation within different intervals of MI, computed using Monte Carlo Integration, to test the conclusion under different settings. The MI value interval is 0.01 and 0.005, respectively.

### A.5 DATA DISTRIBUTIONS

In this part, we provide several plots to visualize the sequences sampled from randomly generated Gaussian mixture distributions used for training.

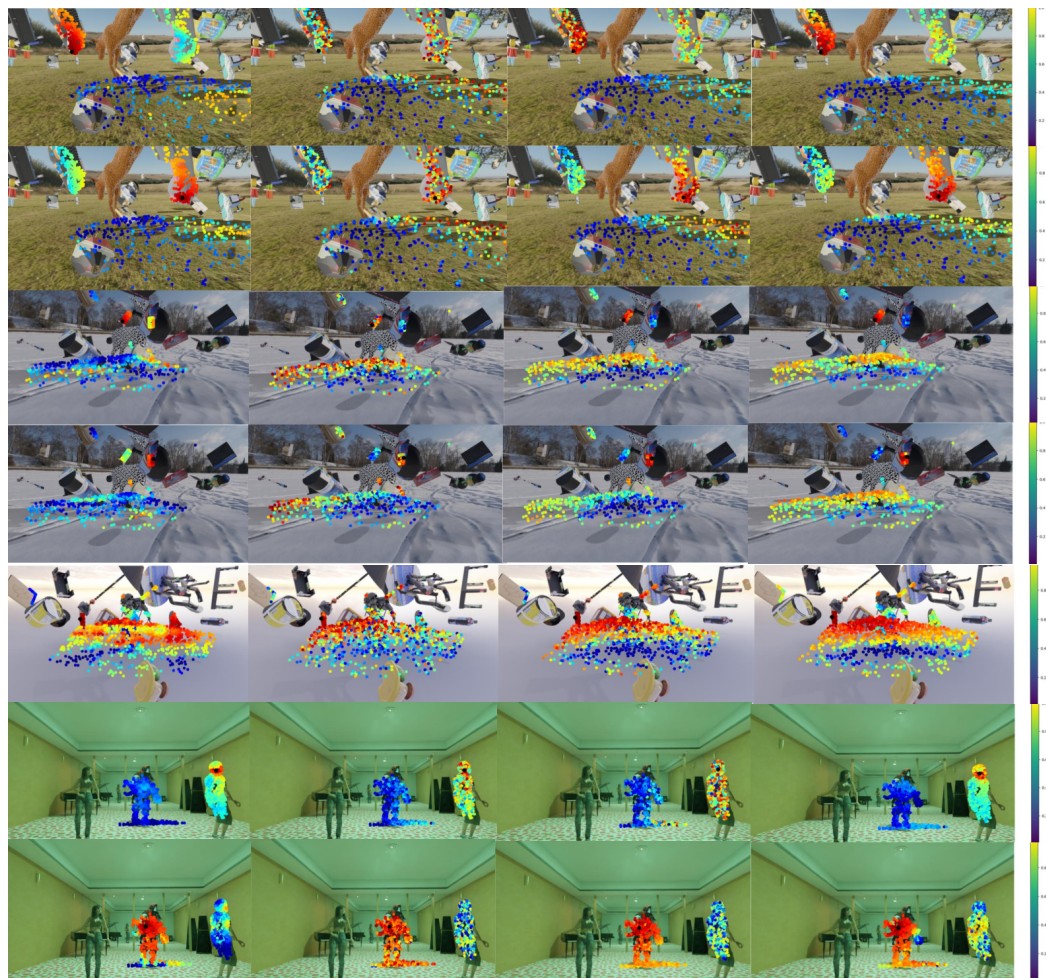

Figure 8: Visual comparison between our model and MINE on the video datasets. They are arranged in the order of InfoNet (no test-time optimization, feed-forward prediction), MINE (iteration 10), MINE (iteration 50), and MINE (iteration 3000) from left to right.

Table 5: Comparison of error mean and variance for given MI value within the range of $\pm 0.005$

|  | MI | 0.0 | 0.1 | 0.2 | 0.3 | 0.4 | 0.5 | 0.6 | 0.7 | 0.8 | 0.9 |
|---|---|---|---|---|---|---|---|---|---|---|---|
| Mean | KSG | 3e-4 | 0.002 | 0.003 | 0.006 | 0.009 | 0.007 | 0.011 | 0.014 | 0.015 | 0.017 |
|  | KDE | 0.008 | 0.010 | -0.003 | -0.038 | -0.070 | -0.114 | -0.158 | -0.197 | -0.249 | -0.295 |
|  | MINE/500 | -0.001 | -0.058 | -0.117 | -0.175 | -0.229 | -0.292 | -0.344 | -0.398 | -0.460 | -0.490 |
|  | MINE/100 | -0.003 | -0.091 | -0.171 | -0.250 | -0.336 | -0.420 | -0.502 | -0.589 | -0.667 | -0.75 |
|  | Ours | **0.015** | **0.011** | **1e-5** | **-0.026** | **-0.056** | **-0.093** | **-0.125** | **-0.154** | **-0.207** | **-0.229** |
| Variance | KSG | 2e-4 | 3e-4 | 4e-4 | 5e-4 | 6e-4 | 7e-4 | 9e-4 | 9e-4 | 1e-3 | 1e-3 |
|  | KDE | 0.008 | 0.005 | 0.004 | 0.003 | 0.006 | 0.009 | 0.011 | 0.015 | 0.017 | 0.019 |
|  | MINE/500 | 4e-6 | 0.001 | 0.004 | 0.007 | 0.014 | 0.019 | 0.029 | 0.040 | 0.051 | 0.056 |
|  | MINE/100 | 1e-5 | 4e-4 | 0.002 | 0.005 | 0.008 | 0.013 | 0.176 | 0.255 | 0.290 | 0.367 |
|  | Ours | **2e-4** | **0.001** | **0.001** | **0.002** | **0.005** | **0.007** | **0.010** | **0.014** | **0.020** | **0.025** |

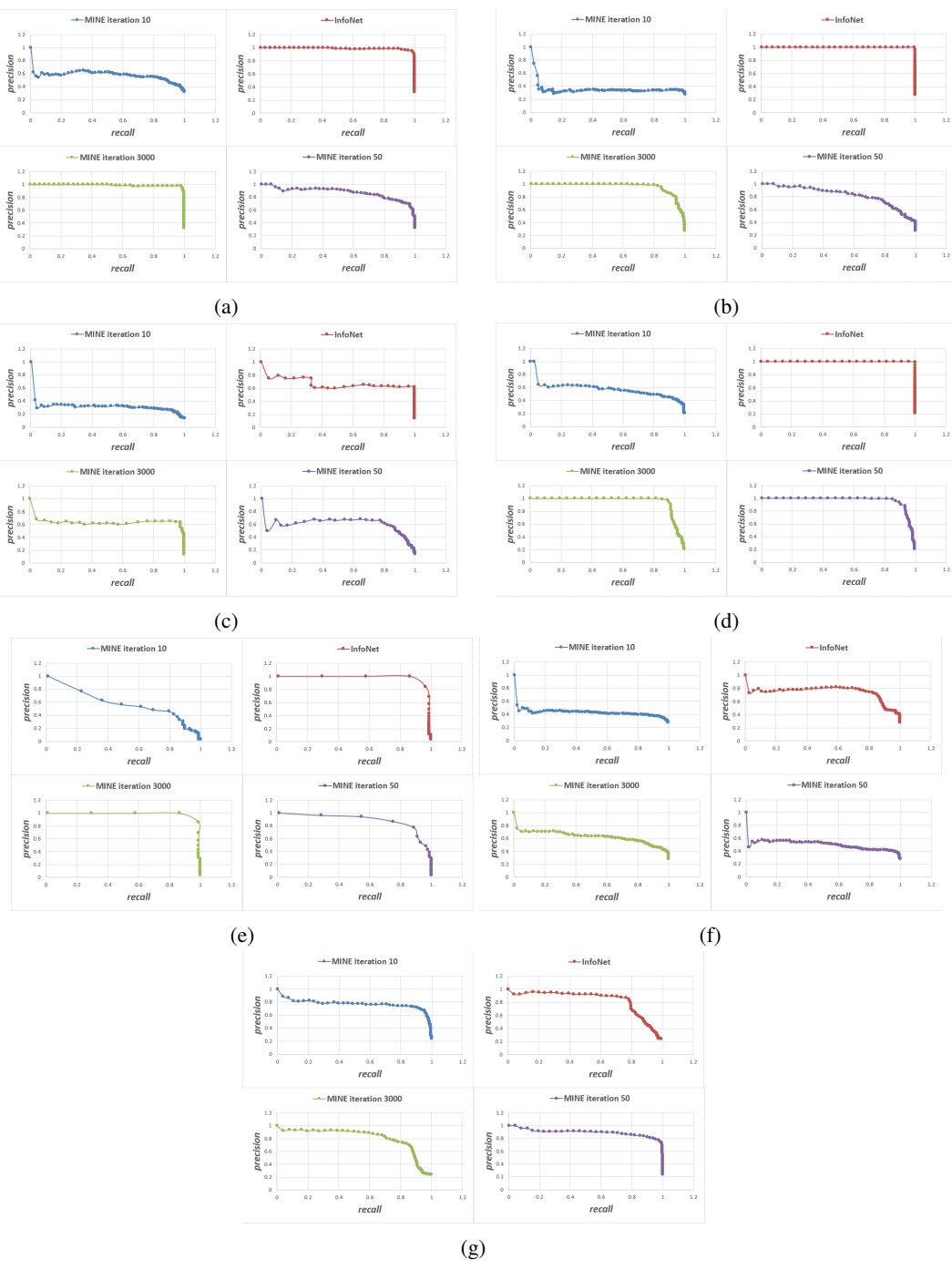

Figure 9: Individual PR graph of our model and MINE. In the experiments conducted on video datasets, InfoNet exhibited notably high stability compared to MINE.

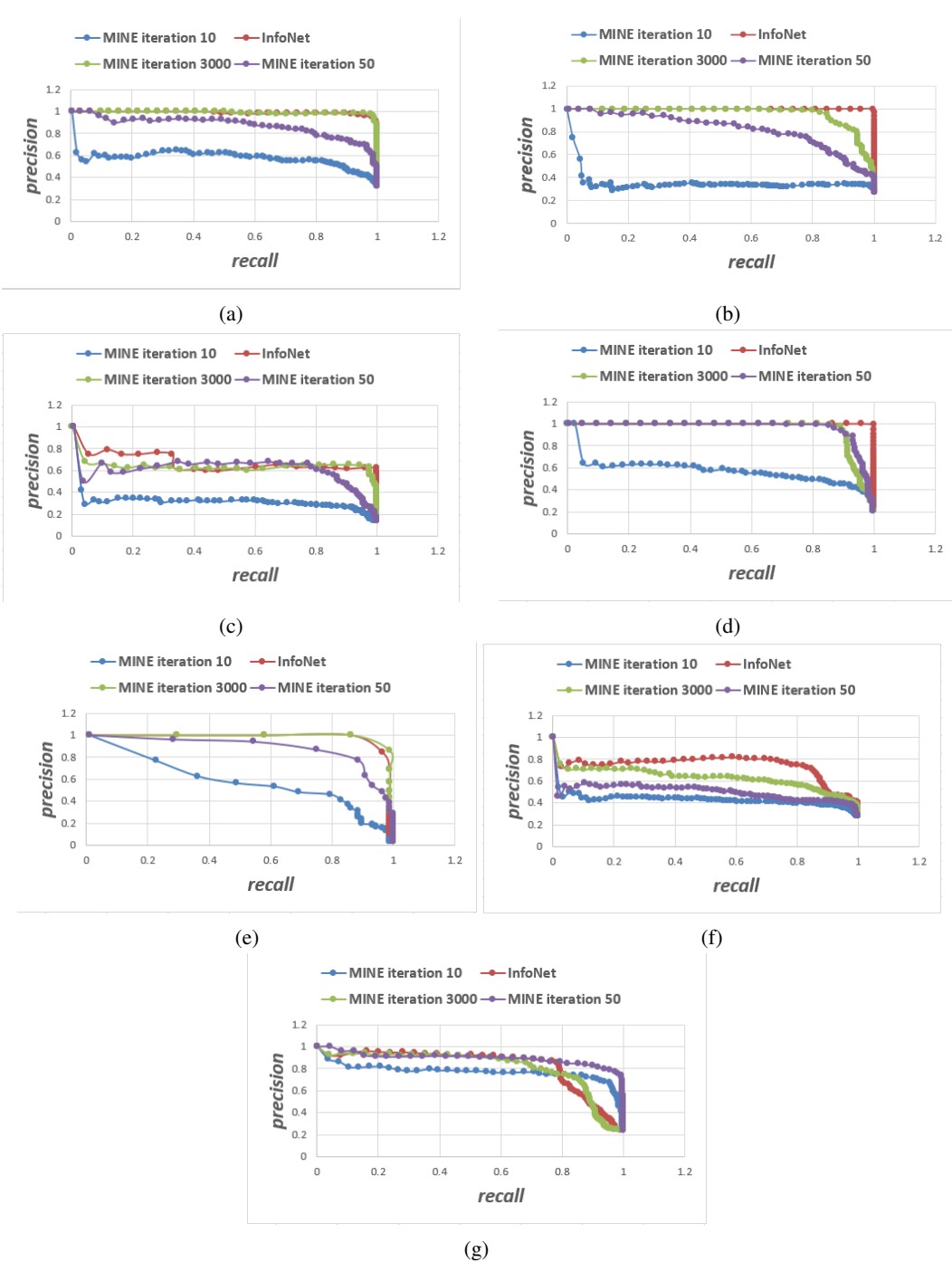

Figure 10: Comparison between PR graphs of our model and MINE. In the same video dataset, InfoNet consistently exhibits superior performance compared to MINE.

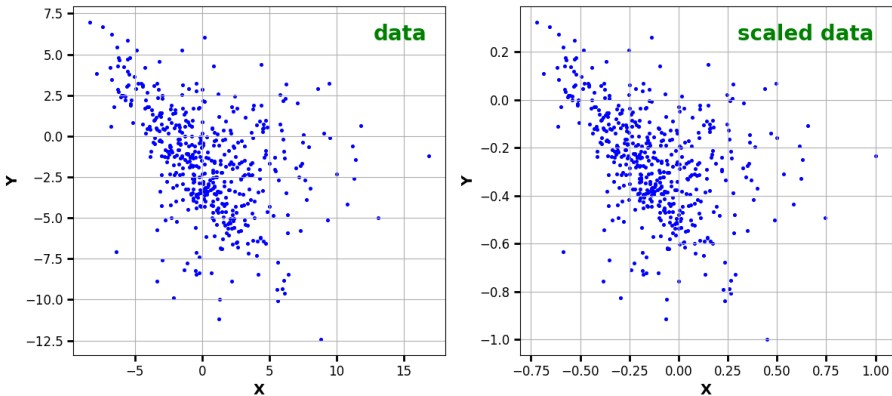

Figure 11: Data points sampled from one mog distribution with 3 components, MI between X and Y is 0.316.

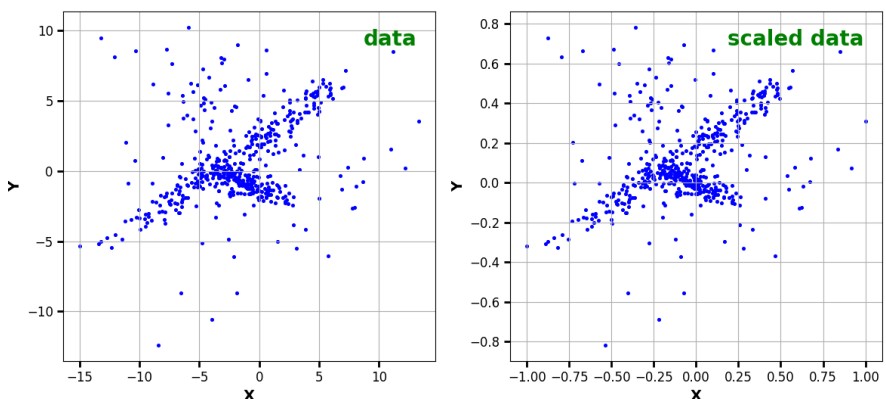

Figure 12: Data points sampled from one mog distribution with 7 components, MI between X and Y is 0.510.

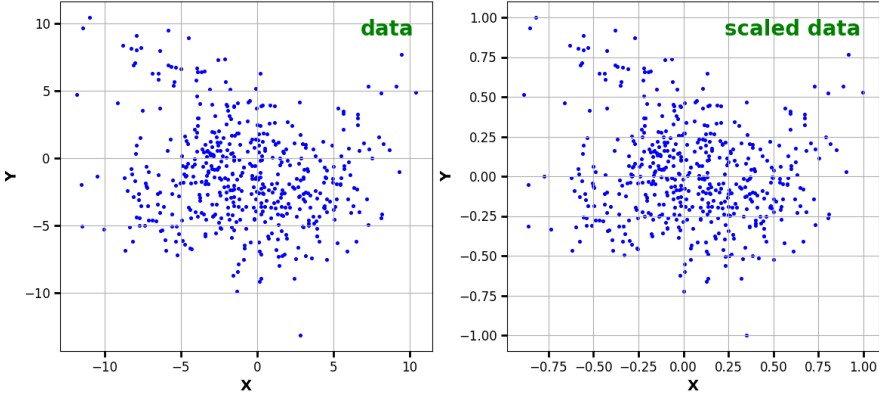

Figure 13: Data points sampled from one mog distribution with 10 components, MI between X and Y is 0.071.

# B  REVISION

## B.1  COPULAS

To enhance the efficiency of mutual information estimation, we introduce the method called copula during the data preprocessing stage. This approach is initiated based on a fundamental property of mutual information: given that $f, g : \mathbb{R} \to \mathbb{R}$ are arbitrary strictly increasing functions, the following equation holds true:

$$\mathbb{I}\left(f(X), g(Y)\right) = \mathbb{I}\left(X, Y\right). \tag{7}$$

Specifically, drawing inspiration from the research presented in (Pál et al., 2010), we select the mappings $f = F_X$ and $g = F_Y$, where $F_X$ and $F_Y$ represent the cumulative distribution functions (CDF) of random variables $X$ and $Y$ respectively. When $F_X$ and $F_Y$ are continuous, the marginal distribution becomes a uniform distribution over the interval $[0, 1]$.

While the specific CDF of $X$ and $Y$ is not known in our situations, we employ the empirical CDF $\left(\widehat{F}_X, \widehat{F}_Y\right)$ as an alternative. Given a sequence $\mathbf{X} = (X_1, X_2, \cdots, X_n)$ with length $n$, where each sample $X_i$, $i = 1, \cdots, n$, originates from an unknown distribution, the empirical CDF is defined as follows:

$$\widehat{F}_X(x) = \frac{1}{n} \operatorname{card}\left(\{i : 1 \leq i \leq n, x \leq X_i\}\right), \quad x \in \mathbb{R}, \tag{8}$$

where $\operatorname{card}(\cdot)$ denotes the cardinality of the set. Note that while $\widehat{F}_X$ does not establish a bijection between $\mathbb{R}$ and the interval $[0, 1]$, it is quite straightforward to create a bijection through interpolation while preserving the order of the sampling points. This ensures the invariance of mutual information, as the property remains unaffected by the transformation.

Our approach involves using the empirical CDF of $X$ and $Y$ to map them to a uniform distribution between $[0, 1]$ prior to training and evaluation. In practice, this mapping process can be reduced to a simple sorting step:

$$f(x) = \frac{1}{n} \operatorname{card}\left(\{i : 1 \leq i \leq n, x \leq X_i\}\right), \quad x = X_1, X_2, \cdots, X_n, \tag{9}$$

and

$$g(y) = \frac{1}{n} \operatorname{card}\left(\{i : 1 \leq i \leq n, y \leq Y_i\}\right), \quad y = Y_1, Y_2, \cdots, Y_n, \tag{10}$$

which are strictly increasing mappings that satisfies the requirement stated in equation 7.

This significantly reduces data complexity, leading to substantial improvements in our model's performance and a considerable increase in the speed of convergence. Since using the copula let the data points fall more evenly in the lookup table. Specifically, the training latency reduction is up to 5 times compared to the previous scheme without incorporating copulas method. By introducing the copula, our latest results are presented with a mark of "(new)".

Figure 14 illustrates the performance of InfoNet on different number of gauss components, all the results are gained by randomly generate 200 pairs of GMM distributions and sort them by the value of real mutual information. We can see that our estimation is close to the ground truth.

## B.2  PERFORMANCE ON OTHER DISTRIBUTIONS

Paper Czyż et al. (2023) provides a diverse family of distributions with known ground-truth mutual information. We select three one-dimension distributions to test our InfoNet performance, note that our model has been only trained on GMM distributions and without any additional training.

**Half-Cube Map** Applying the half-cube homeomorphism $h(x) = |x|^{3/2} \operatorname{sign}(x)$ to Gaussian variables $X$ and $Y$, this could lengthen the tail. The transformation does not influence the ground truth value of MI.

**Asinh Mapping** Applying inverse hyperbolic sine function $\operatorname{asinh} x = \log\left(x + \sqrt{1 + x^2}\right)$ to shorten the tails, this transformation does not change the ground truth value of MI.

**Additive Noise** Let independent r.v. $X \sim \operatorname{Uniform}(0, 1)$ and $N \sim \operatorname{Uniform}(-\varepsilon, \varepsilon)$, where $\varepsilon$ is the noise level. For $Y = X + N$, we could derive $\mathbf{I}(X; Y)$ analytically.

Table 6: Error mean and variance of different MI estimators. Methods that do not rely on neural networks are highlighted in Blue, and those leveraging neural networks are colored Green.

| | MI | 0.0 | 0.1 | 0.2 | 0.3 | 0.4 | 0.5 | 0.6 | 0.7 | 0.8 | 0.9 |
|---|---|---|---|---|---|---|---|---|---|---|---|
| Mean | KSG | 0.001 | 0.001 | 0.004 | 0.006 | 0.008 | 0.009 | 0.012 | 0.015 | 0.016 | 0.014 |
| | KDE | 0.005 | 0.010 | -0.003 | -0.350 | -0.071 | -0.109 | -0.155 | -0.199 | -0.239 | -0.292 |
| | MINE-500 | **-0.003** | **-0.058** | -0.116 | -0.173 | -0.228 | -0.294 | -0.344 | -0.399 | -0.431 | -0.485 |
| | MINE-100 | -0.008 | -0.092 | -0.173 | -0.251 | -0.336 | -0.420 | -0.504 | -0.584 | -0.658 | -0.742 |
| | InfoNet | 0.018 | 0.010 | **0.0001** | **-0.026** | **-0.056** | **-0.087** | **-0.125** | **-0.155** | **-0.193** | **-0.233** |
| | InfoNet(new) | 0.010 | 0.004 | **0.008** | **-0.024** | **-0.040** | **-0.063** | **-0.082** | **-0.101** | **-0.124** | **-0.138** |
| Variance | KSG | 2e-4 | 3e-4 | 4e-4 | 5e-4 | 6e-4 | 8e-4 | 9e-4 | 9e-4 | 1e-3 | 1e-3 |
| | KDE | 0.010 | 0.005 | 0.001 | 0.003 | 0.004 | 0.005 | 0.010 | 0.012 | 0.014 | 0.019 |
| | MINE-500 | **4e-5** | 0.001 | 0.004 | 0.008 | 0.013 | 0.018 | 0.027 | 0.039 | 0.052 | 0.060 |
| | MINE-100 | 4e-5 | **5e-4** | 0.002 | 0.005 | 0.009 | 0.012 | 0.017 | 0.025 | 0.033 | 0.040 |
| | InfoNet | 4e-4 | 0.001 | **0.001** | **0.002** | **0.004** | **0.006** | **0.010** | **0.014** | **0.020** | **0.028** |
| | InfoNet(new) | 1e-5 | 1e-4 | **3e-4** | **8e-4** | **0.001** | **0.002** | **0.004** | **0.005** | **0.007** | **0.009** |

Table 7: Correlation order accuracy of different MI estimators. Methods that do not rely on neural networks are highlighted in Blue, and those leveraging neural networks are colored Green.

| No. of Comps. | 1 | 2 | 3 | 4 | 5 | 6 | 7 | 8 | 9 | 10 |
|---|---|---|---|---|---|---|---|---|---|---|
| KSG | 98.7 | 99.0 | 98.2 | 98.0 | 97.9 | 97.7 | 97.6 | 97.5 | 97.0 | 97.3 |
| KDE | 97.4 | 97.7 | 97.9 | 97.5 | 97.9 | 97.8 | 97.0 | 97.4 | 97.4 | 97.4 |
| MINE-500 | **98.5** | 91.2 | 90.8 | 87.2 | 84.5 | 83.7 | 81.2 | 79.6 | 81.3 | 78.1 |
| MINE-100 | 94.6 | 77.1 | 75.4 | 71.6 | 67.5 | 69.4 | 66.5 | 66.3 | 68.7 | 66.4 |
| MINE-10 | 60.9 | 56.1 | 55.1 | 54.3 | 52.4 | 54.9 | 53.7 | 50.4 | 53.1 | 52.5 |
| InfoNet | 97.3 | **96.2** | **97.0** | **97.5** | **97.1** | **97.6** | **97.2** | **97.2** | **97.8** | **97.4** |
| InfoNet(new) | 100 | **99.76** | **99.76** | **99.54** | **99.54** | **99.6** | **99.48** | **99.56** | **99.64** | **99.58** |

Figure 15 shows our result on other distributions inspite of Mixture of Gaussian distributions. Due to the introduce of copula, our model can suit different monotonic transformation well and this is the reason of good estimating of Half-Cube Map and Asinh Mapping. Also, our model performance on Additive noise well, this illustrate our model have good generalization ability since we do not train any uniform distributions and adding noise in the train process.

## B.3 High Dimension Results

Our method indeed encounters specific challenges in high-dimensional settings, primarily due to the inherent constraints associated with simulation-based learning. As the dimensionality of data escalates, capturing a comprehensive array of distribution scenarios becomes increasingly complex. Additionally, the exponential growth in the number of grids within the lookup table, as dimensions rise, makes our proposed discrete representation method less feasible.

To address these issues, we introduce the concept of sliced mutual information (SMI) Goldfeld & Greenewald (2021), defined as the average of mutual information (MI) terms between one-dimensional random projections. Suppose $X$ and $Y$ are $d_x$-dimensional and $d_y$-dimensional random variables. SMI can be expressed as the expected MI of one-dimensional random projections:

$$\text{SMI}(X;Y) = \mathbb{E}_{\phi,\psi}\left[\mathbb{I}(\phi(X);\psi(Y))\right] = \frac{1}{S_{d_x-1}S_{d_y-1}}\oint_{\mathbb{S}^{d_x-1}}\oint_{\mathbb{S}^{d_y-1}}\mathbb{I}\left(\theta^\top X;\phi^\top Y\right)\mathrm{d}\theta\mathrm{d}\phi \quad (11)$$

Here, $\mathbb{S}^{d-1}$ denotes the $d$-dimensional sphere (whose surface area is designated by $S_{d-1}$), $\phi$ and $\psi$ are vectors used for linear projection from high-dimensional space to one-dimensional space, and $\mathbb{E}_{\phi,\psi}$ denotes the expectation over these projection functions.

While SMI typically yields lower values compared to MI, it retains many of the intrinsic properties of MI and exhibits a certain degree of correlation with it. This inter-connectedness is crucial, as it implies that while SMI offers a novel approach to handling high-dimensional data, it still adheres

to the fundamental principles of MI, thereby ensuring consistency in its theoretical foundations and practical applications.

In this section, we rigorously assess our InfoNet model's efficacy in handling high-dimensional data. This evaluation is carried out through experiments encompassing three distinct tasks: the correlation order test using Multivariate Gaussian distributions, an independence test in high-dimensional settings, and a correlation order test employing real-world data generated via the CLIP model.

### B.3.1 Multivariate Gaussian Correlation Order Test

In this section we similarly compare the capability of InfoNet in classify the correct correlation order on $d$-dimensional gauss distributions: $(X, Y) = \left((X^1, X^2, \cdots X^d), (Y^1, Y^2, \cdots Y^d)\right) \sim \mathcal{N}(\boldsymbol{\mu}, \Sigma)$.

This result shows that our InfoNet model reaches high accuracy and still costs low time complexity. Since our model allows parallel computing on multiple GPUs, it can compute massive scalable MI of projected variables in one feed forward process.

Table 8: Correlation order accuracy of different MI estimators. Methods that do not rely on neural networks are highlighted in Blue, and those leveraging neural networks are colored Green. MINE-100 means training MINE method for 100 iterations, InfoNet-100 means we do 100 times random projection to get an average.

| Dimensions | 2 | 3 | 4 | 5 | 6 | 7 | 8 | 9 | 10 |
|---|---|---|---|---|---|---|---|---|---|
| KSG | 94.4 | 95.5 | 91.8 | 92 | 94.1 | 93.6 | 94.1 | 94.1 | 94.2 |
| Energy Distance | 49.6 | 51.2 | 52.2 | 51.5 | 52.5 | 49.6 | 48.7 | 50.2 | 51.3 |
| MINE-100 | 78.5 | 82.1 | 86.7 | 84.7 | 88.4 | 89.8 | 90.1 | 90.4 | 90 |
| MINE-1000 | 93.6 | 93.9 | 94.4 | 94.3 | 91.6 | 91.7 | 89.5 | 91 | 90.3 |
| MINE-5000 | 96.2 | 97 | 97 | 96.2 | 94.9 | 94.2 | 93.2 | 92.8 | 93 |
| InfoNet-100 | 93.7 | 94.6 | 94.4 | 95.7 | 93.3 | 95.8 | 95.8 | 95.4 | 93.8 |
| InfoNet-500 | 94.9 | 93.7 | 95.7 | 95.8 | 97.1 | 96.4 | 97.2 | 97.8 | 96.8 |
| InfoNet-1500 | **97.7** | **96.4** | **96.2** | **97.9** | **97.4** | **98.1** | **98.2** | **97.3** | **98.3** |

Table 8 shows that our method is good at classifying the correct order compares to MINE, KSG and Energy Distance in Rizzo & Székely (2016).

### B.3.2 Independence Testing

In this part we check the performance of our model on high dimensional independence testing, due to the fact that $\text{SMI}(X, Y) = 0$, $\text{MI}(X, Y) = 0$, $\text{ED}(p_{X,Y}(x, y), (p_X(x), p_Y(y)) = 0$ if and only if $X$ and $Y$ is independent, where ED represents Energy Distance Rizzo & Székely (2016)

Figure 16 shows independence testing results for three relationships between $X, Y$ pairs. The figure shows the area under the curve (AUC) of the receiver operating characteristic (ROC) for independence testing via our methods (using sliced mutual information) and MINE along with Energy Distance. the random projection steps for SMI uses 1000 random slices, and the ROC curves are computed from 100 random trials . Figure uses sample size selected from each distribution $n$ as variable to show the AUC value under different sample sizes, and the dimension $d$ we test is 16, 128. The joint distribution of $(X, Y)$ in each case of is:

(a) **One feature (linear)**: $X, Z \sim \mathcal{N}(0, \mathrm{I}_d)$ i.i.d. and $Y = \frac{1}{\sqrt{2}}\left(\frac{1}{\sqrt{d}}\left(\mathbf{1}^\top X\right)\mathbf{1} + Z\right)$, where $\mathbf{1} := (1, \ldots, 1)^\top \in \mathbb{R}^d$.

(b) **Two features**: $X, Z \sim \mathcal{N}(0, \mathrm{I}_d)$ i.i.d. and $Y_i = \frac{1}{\sqrt{2}}\begin{cases} \frac{1}{d}\left(\mathbf{1}_{\lfloor d/2 \rfloor} 0 \ldots 0\right)^\top X + Z_i, & i \le \frac{d}{2} \\ \frac{1}{d}\left(0 \ldots 0 \mathbf{1}_{\lceil d/2 \rceil}\right)^\top X + Z_i, & i > \frac{d}{2}. \end{cases}$

(c) **Independent coordinates**: $X, Z \sim \mathcal{N}(0, \mathrm{I}_d)$ i.i.d. and $Y = \frac{1}{\sqrt{2}}(X + Z)$.

### B.3.3 Correlation Order Test Using Data Generated by CLIP

In this section we check the correlation order performance on high dimension data generated by CLIP Radford et al. (2021).

The task is the correlation order prediction, which means, given three random variables, X,Y, and Z, we would like to find which one of Y and Z is more correlated with X.

Specifically, we download the ImageNet2017(ILSVRC) dataset and CLIP pre-trained model. Denote the image encoder of CLIP as a function $f_{\text{CLIP}}$, which takes in an image and outputs a 512-dimension vector representing the feature of the image. We begin with a randomly selected sequence of images from ILSVRC dataset. Then $X$ is generated by $X = f_{\text{CLIP}}(\text{image})$ to be a sequence of 512-dimensional vectors. Then we add Gaussian noise to the original image to get $Y$: $Y = f_{\text{CLIP}}(\text{image} + \text{noise})$. Finally, we add color jitter to randomly change the image's brightness, contrast, saturation, and hue. We denote the transformation as $g_{jitter}$. Then $Z$ is generated by $Z = f_{\text{CLIP}}(g_{\text{jitter}}(\text{image} + \text{noise}))$. In this case, we can tentatively ensure that the ground truth is $MI(X,Y) > MI(X,Z)$ as the information contained in the image for getting Z is less than that for getting Y with respect to the original image, according to the Data Processing Inequality.

Table 9: Accuracy Comparison

| Method | Ours | MINE |
|---|---|---|
| accuracy | 100 | 100 |

From table 9 we can see that MINE and our method reaches good consistency. In the experiment we train MINE for 1000 iterations and InfoNet does 1000 random projections. The number of the sequence of images is 5000.

### B.4 Discussion on Simulation-Based Intelligence

The connection between our work on InfoNet and the field of simulation-based intelligence lies in the underlying principles of learning from data and adapting to complex environments. Both approaches aim to improve decision-making and intelligent behavior by leveraging statistical relationships and learning mechanisms.

In the case of InfoNet, the focus is on efficiently estimating mutual information (or HGR maximal correlations) between random variables or data streams, which is crucial for understanding and capturing the dependencies in complex systems. By developing a feed-forward neural estimator for mutual information, our work contributes to improving the efficiency and differentiability of correlation estimation in various applications.

Simulation-based intelligence Ramon et al. (2021) Cranmer et al. (2020), on the other hand, involves AI agents learning and adapting to complex environments through interactions within a simulated setting. These agents often use reinforcement learning or other learning mechanisms to estimate statistical relationships and correlations in order to make better decisions and improve their performance.

Both fields share a common goal of enhancing intelligent behavior and decision-making by using data-driven approaches. The efficient mutual information (or HGR maximal correlation) estimation provided by InfoNet can potentially benefit simulation-based intelligence systems by allowing them to better understand the relationships between variables in their environment. This improved understanding can lead to more effective learning and adaptation, ultimately enhancing the overall performance of AI agents in simulation-based intelligence applications.

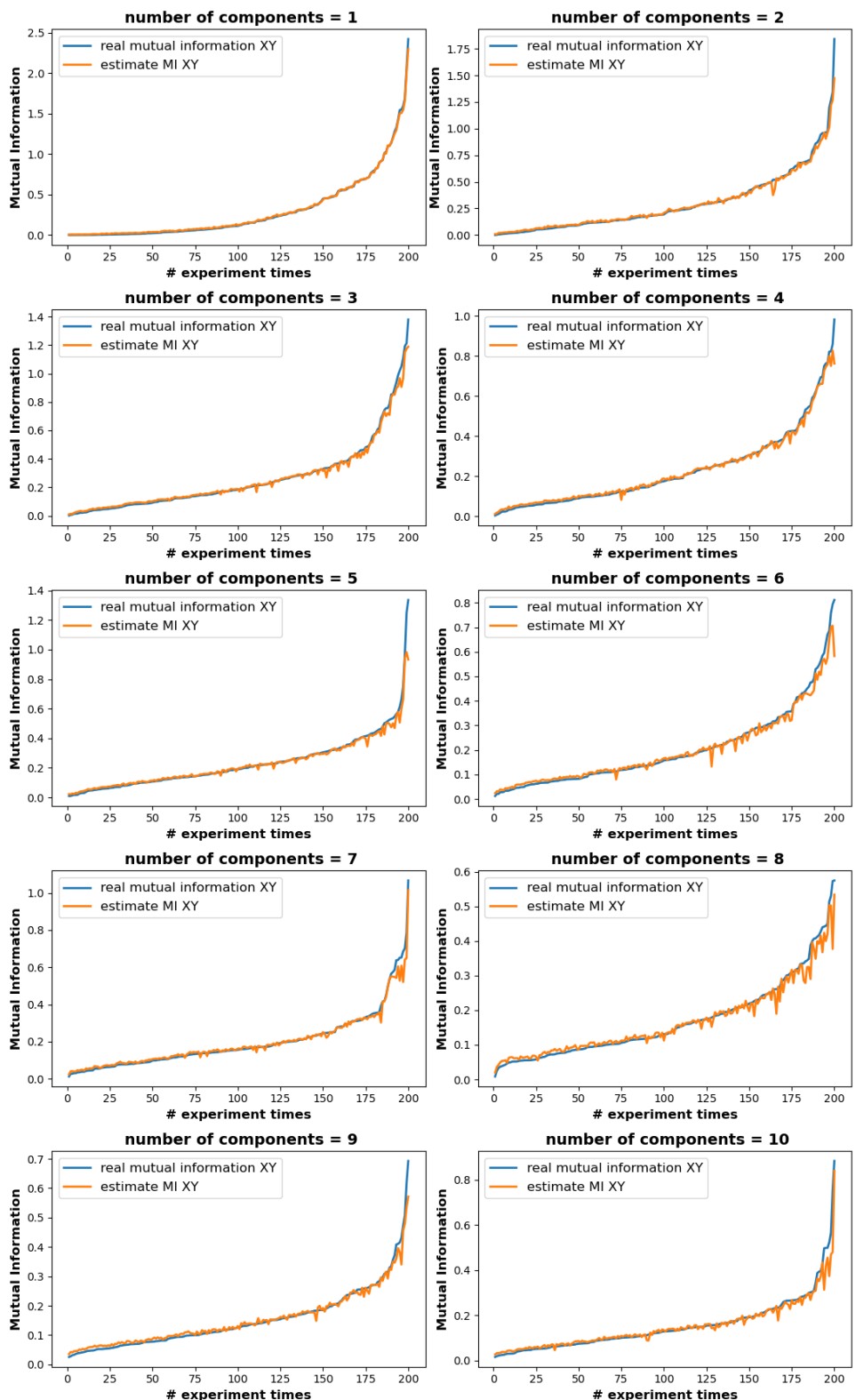

Figure 14: This figure shows our InfoNet performance on different number of gauss components, all the results are gained by randomly generate 200 pairs of distributions and sort them by the value of real mutual information. We can see that our work can reach good performance on scalable data.

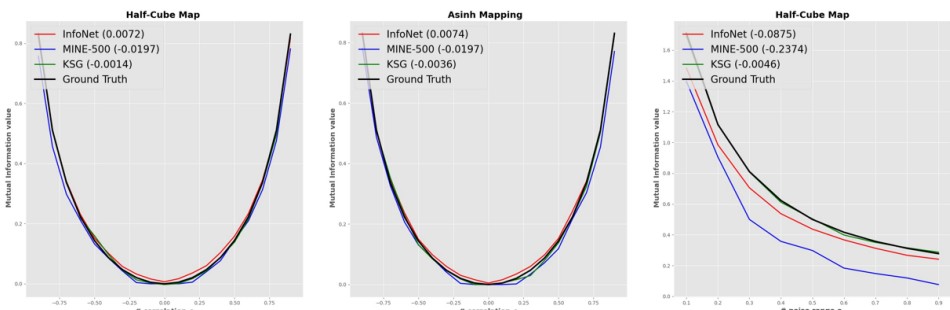

Figure 15: Evaluation of performance on distribution other than GMM, comparing with MINE with 500 training iterations and KSG with nearest neighbour number $k = 1$. From the figure we can see that our InfoNet model only trained on GMM data suits other distributions well and the result is close to the ground truth.

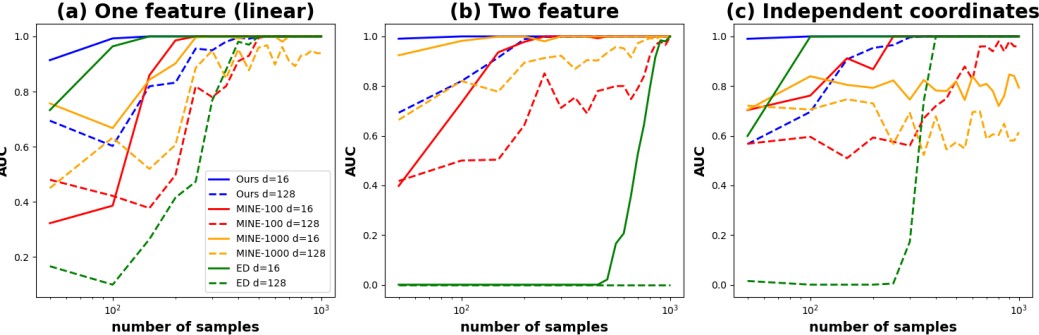

Figure 16: Independence testing over InfoNet, MINE, ED. Figure shows the area under the curve (AUC) of the receiver operating characteristic (ROC) as a function of sample n.

