# OpenReview forum: "InfoNet: An Efficient Feed-Forward Neural Estimator for Mutual Information"
_ICLR.cc/2024/Conference — Submitted to ICLR 2024_

### Official Review · Reviewer_zWhA · 2023-10-20

**Soundness:** 3 good
**Presentation:** 3 good
**Contribution:** 3 good
**Rating:** 6
**Confidence:** 4

**Summary:**

This works present a new method for real-time mutual information (MI) estimate. The core of the method is a network that maps a population of empirical samples to the optimal function r(x,y) = p(x, y)/p(x)p(y). This optimal function is represented by a look-up table where the values of x, y are quantized into several bins (e.g. 100). To train this network, the authors use an idea similar to simulation-based inference, that they first extensively simulate data from various artificial distributions, then learn the relationship between the data population and the function r(x, y) behind. It is believed that after seeing sufficiently many artificial distributions, the network can cover the case in real world. Compared to traditional methods which learns r(x, y) from scratch, this method pre-compute r(x, y) via simulation, thereby is much faster at inference time. In addition to MI estimate, the idea can also be extended to compute Renyi's maximal correlation coefficient.

**Strengths:**

- **Significance**: the problem of scalable estimation of mutual information and statistical dependence is very fundamental in modern data science and machine learning, and by far there is still no method that can be both accurate (like neural network-based method) and efficient (like non-parameteric method). The method does a good job in trying to take the advantage of both worlds;
- **Novelty**: the idea of this work, to my knowledge, is very novel within the context of MI and statistical dependence estimate. While many recent works have also focused on scalable computation of mutual information and statistical dependence [1, 2, 3, 4], this work presents an orthogonal possibility. The idea of learning to map the data population to the optimal neural estimator via massive simulation is so interesting, and the way to implement this idea via attention-based network is also technically sensible. In fact, the idea of simulation-based learning is not new, particularly in the field of statistics (see e.g. [5, 6]). However, the paper is the first one to apply this idea to statistical dependence modelling;
- **Presentation**: the paper is also mostly well-written and easy to follow. I enjoy reading the paper very much.

References:

[1] The Randomized Dependence Coefficient, NeurIPS 2017

[2] Sliced mutual information: A scalable measure of statistical dependence, NeurIPS 2021

[3] Scalable Infomin Learning, NeurIPS 2022

[4] Fairness-Aware Learning for Continuous Attributes and Treatments, ICML 2019

[5] Simulation intelligence: Towards a new generation of scientific methods, arxiv 2112.03235

[6] The frontier of simulation-based inference, PNAS 2020

**Weaknesses:**

- **Limitations** (major): while very interesting, the proposed method seems only work for low-dimensional data. This is due to (a) the limitation of simulation-based learning itself, where as the dimensionality grows it becomes more and more difficult to cover diverse cases of distributions; (b) the number of grids in the lookup table grows exponentially with the dimensionality of data, making the discrete representation proposed impractical. That being said, the idea can expect to work well in e.g. 1D/2D cases;
- **Experiments** (major):  the experiments in this paper are done on synthetic dataset and one (relatively simple) real-world dataset, which is a bit disappointing compared to the interesting idea. In this regard, I think the current experiments do not really show the potential of the method developed. The synthetic task regarding maximal correlation is also not so intutitive/easy-to-understand to me compared to those done in existing works [1, 3, 4]. Why not test the power of the method in the presence of different statistical association patterns (see [1, 3, 4])?;
- **Discussion on relationship to related works**: as mentioned in the strength block above, the paper presents a new method for estimating statistical dependence in a highly efficient way. It will be ideal to have a discussion between your method and these existing works, highlighting why your method is more preferable compared to these prior works. In addition, it *may* also be nice to mention the connection of your work and the field of simulation-based intelligence due to the inherent similarity between the underlying principles. This will connect your work with the broader community;
- **Presentation** (minor): while the presentation is overall good, it was only after reading Section 3.1 and Algorithm 1 I can understand your method. In fact, Section 3.1 seems to be important description of the core training algorithm of your method. I think it would be easier to understand for readers if the authors make this part a separate section (e.g. training algorithm) rather than part of the experiments. There are also some typos/errors in the text (for example `MIN-100’ in Table 1).

**Questions:**

- Is it really necessary to parameterize the network to be a lookup table ? Why can’t we e.g. directly output the MI/Maximal correlation value for the input data population?
- In the experiment, why study cases with known ground-truth maximal correlation? For example, you can generate data y = f(x) + eps with known f and calculate the maximal correlation with f.

---

> ### Author Response · Authors · 2023-11-21
>
> Dear Reviewer zWhA,
>
> Thanks for your comprehensive review and the depth of your engagement with our manuscript, and the references you provide give us a lot of inspiration. Your acknowledgment of our method's significance and novelty within the field of mutual information is immensely encouraging. We are also grateful for your constructive critique, which has prompted an expansion of the method's scalability and experimental validation.
>
> In response to your comments and suggestions, we have expanded our experimental scope and provided clarifications on the implementation details in our revised manuscript, which we believe can help reflect the potential and applicability of our proposed approach in a more accurate manner.
>
> We hope that these clarifications can help finalize your assessment and the rating of our paper. Please also let us know if you have any further questions that we need to provide additional clarifications.
>
> ***
>
> **W1: ``Limitations (major): while very interesting, the proposed method seems only work for low-dimensional data. This is due to (a) the limitation of simulation-based learning itself, where as the dimensionality grows it becomes more and more difficult to cover diverse cases of distributions; (b) the number of grids in the lookup table grows exponentially with the dimensionality of data, making the discrete representation proposed impractical. That being said, the idea can expect to work well in e.g. 1D/2D cases;''**
>
>
>
> **A**: Thank you for your insight into the difficulty of simulation/sampling in high-dimensional space, which may increase the size of the look-up table exponentially.
>
> In the following, we would like to provide an alternative for high-dimensional MI estimation using the pre-trained InfoNet with the sliced technique provided in [1], which projects a high-dimensional signal to a low dimension and allows efficient MI estimation.
>
> We have rigorously evaluated our InfoNet model on high-dimensional random variables compared with MINE, in terms of correlation order accuracy and independence test.
>
> Specifically, we validated the pretrained InfoNet in two tasks:
>
> Task 1: given three jointly sampled random variables, X, Y, and Z from high-dimensional Gaussian, we want to know which one of Y and Z is more correlated with X, where the correct order, i.e., MI(X,Y)>=MI(X,Z) or MI(X,Y)<MI(X,Z)  is obtained by the ground-truth MI computed with the analytical solution of MI for Gaussian. The performance metric is the binary classification accuracy.
>
> Task 2: Independence test. Given a pair of random variables X, and Y, either independent or not, we apply InfoNet and MINE to check their independence since both metrics can claim independence between two random variables if the metrics are close to zero. We vary the epsilon as the threshold, and we plot the precision-recall curve to compare their performance using the area under the curve (AUC, the closer to 1, the better).
>
> Results:
>
> A. In terms of the correlation order test (Task 1), we get the following performance:
>
> | Dimensions | 2    | 3    | 4    | 5    | 6    | 7    | 8    | 9    | 10   |
> |------------|------|------|------|------|------|------|------|------|------|
> | MINE-5000  | 96.2 | 97   | 97   | 96.2 | 94.9 | 94.2 | 93.2 | 92.8 | 93   |
> | InfoNet | **97.7** | **96.4** | **96.2** | **97.9** | **97.4** | **98.1** | **98.2** | **97.3** | **98.3** |
>
> From the table, we can see that InfoNet performs better in terms of estimating the correlation orders between a triplet of random variables across multiple dimensions.
> Moreover, InfoNet runs at 0.05s per estimate, while MINE runs at 12.3s per estimate.
>
> B. Results of AUC on independence testing (Task 2)
> | Sequence length | 50    | 100    | 150    | 200    | 250    | 300    | 350    | 400
> |------------|------|------|------|------|------|------|------|------|
> | MINE-1000 d=16 | 0.757 | 0.668 | 0.843 | 0.902 | 0.996 | 1. | 1 | 1 |
> | MINE-1000 d=128 | 0.45 | 0.633 | 0.520 | 0.607 | 0.886 | 0.949 | 0.848 | 0.954 |
> | InfoNet d=16  | 0.914 | 0.992   | 1.0   | 1.0 | 1.0 | 1.0 | 0.99 | 1.0 |
> | InfoNet d=128  | 0.694 | 0.603   | 0.819   | 0.832 | 0.954 | 0.950 | 0.981 | 0.992 |
>
> From the table, we can see that in both 16- and 128-dimensional space, InfoNet with the sliced technique can achieve better AUC for independence testing, which shows the potential of the proposed in high-dimensional applications even only trained on low-dimensional signals.
> Details of this experiment can be seen in B.4.1 of the revised version.
>
> [1] Sliced mutual information: A scalable measure of statistical dependence, NeurIPS 2021

---

> > ### Author Response · Authors · 2023-11-21
> >
> > **W2: ``Experiments (major): the experiments in this paper are done on synthetic dataset and one (relatively simple) real-world dataset, which is a bit disappointing compared to the interesting idea. In this regard, I think the current experiments do not really show the potential of the method developed. The synthetic task regarding maximal correlation is also not so intutitive/easy-to-understand to me compared to those done in existing works [1, 3, 4]. Why not test the power of the method in the presence of different statistical association patterns (see [1, 3, 4])?;''**
> >
> > **A**: Thank you for your valuable feedback. In response, we have expanded our experiments to include a comprehensive array of tests, including tests on more diverse distributions/tasks and real-world data.
> >
> > First, we rigorously evaluated our InfoNet, which is trained exclusively on Gaussian Mixture Model (GMM) data, against a variety of non-Gaussian mixtures with known ground truth mutual information. The outcomes of these tests were remarkably close to the actual ground truth values, demonstrating the generalization capabilities of InfoNet. For a detailed exploration of these experiments, please refer to Appendix B.2.
> >
> > Also, thank you for the references [1,3,4]. We realize that the referenced studies extensively test independence across various distributions. Similarly, we have conducted independence testing on high-dimensional distributions, testing three different statistical association tasks, which are evaluated by the area under the curve (AUC) metric ([2]). The results are presented in Section B.4.1.
> >
> > Besides, we add another evaluation task to show that the InfoNet model is capable of handling high-dimensional real data. The task is the correlation order prediction, which means, given three random variables, X,Y, and Z, we would like to find which one of Y and Z is more correlated with X. However, it is performed with high-dimensional features from real images.
> >
> > Specifically, we download the ImageNet2017(ILSVRC) dataset and CLIP pre-trained model. Denote the image encoder of CLIP as a function $f_{CLIP}$, which takes in an image and outputs a 512-dimension vector representing the feature of the image. We begin with a randomly selected sequence of images from ILSVRC dataset. Then $X$ is generated by $X=f_{CLIP}(image)$ to be a sequence of 512-dimensional vectors. Then we add Gaussian noise to the original image to get $Y$: $Y = f_{CLIP}(image + noise)$. Finally, we add color jitter to randomly change the image's brightness, contrast, saturation, and hue. We denote the transformation as $g_{jitter}$. Then $Z$ is generated by $Z=f_{CLIP}(g_{jitter}(image+noise))$. In this case, we can tentatively ensure that the ground truth is $MI(X,Y)>MI(X,Z)$ as the information contained in the image for getting Z is less than that for getting Y with respect to the original image, according to the Data Processing Inequality.
> >
> > Accuracy compare.
> > | Method | Ours | MINE-1000 |
> > |------------|----|----|
> > | accuracy  | 100 | 100 |
> >
> > We apply our InfoNet model on this task using sliced mutual information to test the order and reach 100% accuracy (in consistency with MINE). It shows that our method can generalize to real-world data in high dimensions, with high efficiency, e.g., InfoNet runs at 0.05s per estimate, and MINE runs at 2.8s per estimate.

---

> > > ### Author Response · Authors · 2023-11-21
> > >
> > > **W3: ``Discussion on relationship to related works: as mentioned in the strength block above, the paper presents a new method for estimating statistical dependence in a highly efficient way. It will be ideal to have a discussion between your method and these existing works, highlighting why your method is more preferable compared to these prior works. In addition, it may also be nice to mention the connection of your work and the field of simulation-based intelligence due to the inherent similarity between the underlying principles. This will connect your work with the broader community;''**
> > >
> > >
> > >
> > > **A**: Thanks for your nice comment. We will have a more detailed discussion on this point in the final version of our paper. The proposed InfoNet not only offers enhanced efficiency compared to conventional neural-based approaches but also features the key advantage of differentiability. This attribute ensures that the proposed InfoNet can be seamlessly incorporated into a variety of neural network training methods.
> > >
> > > We greatly appreciate your suggestion to connect our work with simulation-based intelligence. In response, we have included a discussion (see B.4 in the revised paper) on how our method aligns with the principles of simulation-based intelligence. This addition will be incorporated into the final version of our paper.
> > >
> > >
> > > ***
> > >
> > >
> > > **W4:`` I think it would be easier to understand for readers if the authors make this part a separate section (e.g. training algorithm) rather than part of the experiments. There are also some typos/errors in the text (for example `MIN-100’ in Table 1).''**
> > >
> > > **A**: Thank you for your suggestions. We have incorporated them in the updated paper, e.g., we make the training detail an independent section, ensuring clarity.
> > >
> > > ***
> > >
> > > **Q1: ``Is it really necessary to parameterize the network to be a lookup table ? Why can’t we e.g. directly output the MI/Maximal correlation value for the input data population?''**
> > >
> > >
> > > **A1**: Thanks for the question. Theoretically, it's possible to bypass the use of a lookup table. Actually, in the initial stage of the development, we have explored a model directly outputting the corresponding \( \theta \) values. However, during training, we encountered an unexpected trend where the model consistently failed to optimize the Donsker-Varadhan formula and output zeros for all distributions. Later after we incorporated a lookup table in the architecture, the problem of non-convergence was gone. We would like to say that the lookup table is beneficial from our experience, though limited since we have not explored the whole space of different architecture.
> > >
> > > Moreover, the lookup table enables efficient computation of \( \theta(x_{\text{new}}, y_{\text{new}}) \) for new samples, e.g., by just performing interpolation without re-running the network, thereby enhancing online processing efficiency.
> > >
> > >
> > > ***
> > >
> > >
> > > **Q2:``In the experiment, why (not) study cases with known ground-truth maximal correlation? For example, you can generate data y = f(x) + eps with known f and calculate the maximal correlation with f.''**
> > >
> > > **A2**: We have conducted an additional experiment according to the suggestion, where \( X \) is derived from a one-dimensional uniform distribution and \( Y = f(X) \), with \( f \) representing five specific functions. In this setup, the MCC's ground truth is 1. We did not add noise to the process so that the results could be evaluated against an accurate ground truth. We plot the error’s mean and variance of the estimated MCC using InfoNet:
> > >
> > > | Function F | X | X^2 | X^3 | X^4 | sin(X) |
> > > |------------|---|-----|-----|-----|--------|
> > > | Ours Mean  | 0 | 0 | 0 | 0 | 0 |
> > > | Ours Var   | 0 | 0 | 0 | 0 | 0 |
> > >
> > > The table above shows that our model is robust and can give an accurate estimate of MCCs between variables under the suggested model (without noise).

---

> ### Comment · Reviewer_zWhA · 2023-11-22
> **Reply to authors**
>
> I thank the authors for their detailed and effective rebuttal. I am largely satisfied with your response. I'll keep my already positive score at this stage.
>
> Below are some further comments:
>
> - **On scaling to higher dimensions with slicing**. I am glad to see that the authors find the literature [2] helpful. Your new idea of utilizing this method to scale to high-dimensional case is indeed sensible and interesting, and the new experiments on higher-dimensional case and real-world data (e.g. CLIP) are helpful. However, as far as I am concerned, this new method might have two problems. First, it no more estimates the original mutual information, meaning that it may not be so useful in cases where the exact value of mutual information is of interests (for example, formally investigation of the trade-offs in data compression/privacy leakage). Second, the required number of slices may grow exponentially as the dimensionality increases [3]. That being said, I think your new method a very sensible extension, and all these above are only remarks rather than critiques.
>
> - **Discussion on flexible/scalable alternatives for measuring statistical dependence**. As mentioned in the review, there also exist other parametric techniques that allow scalable/fast computation of statistical dependence. For example, the statistical dependence metrics in [1, 3, 4] can all be computed analytically (similar to solving linear regression/eigen-decomposition), which are also fast to execute compared to neural network training but are still much more powerful than non-parametric/kernel-based methods. Importantly, these methods are also fully differentiable like your method, so they can be applied to representation learning (see e.g. [3]). When I read your paper, I had a sense that why not using these already proven approaches, which are tested on high-dimensional cases (see e.g. [3]) and cases with diverse association patterns (see e.g. [1, 4]). I think it would be highly helpful to include a short discussion between your method and these methods, highlighting that your idea (pre-training) is orthogonal to these methods, and is conceptually more efficient and accurate. I believe this will help readers better understand the merit of your method.
>
> Other comments: Algorithm 1 and Algorithm 2 seem duplicated?
>
> [1] The Randomized Dependence Coefficient, NeurIPS 2017
>
> [2] Sliced mutual information: A scalable measure of statistical dependence, NeurIPS 2021
>
> [3] Scalable Infomin Learning, NeurIPS 2022
>
> [4] Fairness-Aware Learning for Continuous Attributes and Treatments, ICML 2019

---

> ### Author Response · Authors · 2023-11-22
>
> Dear Reviewer zWhA,
>
>
> Thanks a lot for your positive response. We are grateful for your insightful comments and appreciation of our new experiments. We will incorporate them into our final version and handle the discussion among the mentioned references carefully and with clarity. Also, thanks for pointing out the duplication of the algorithms. We intentionally keep them here so the change is trackable, but will definitely merge them in the final version.
>
> Also, in case it may cause curiosity on the number of projections for the sliced technique we have used, we provide them for each dimension below:
>
>
> | Dim  |     8   |   16  |   128 |   512 |
> |---------|-----|-----|-----|-----|
> |#proj.  | 100 | 100 |  500 |   500 |
>
>
> These numbers (of projections) are typical and can guarantee reasonable performance in our experiments. We also have an ablation on the performance improvement over different numbers of projections in Table 8. Moreover, due to the fact that our network can handle different projections in batch mode, the prediction efficiency is still maintained.
>
> Thanks again for your valuable feedback！

---

> ### Comment · Reviewer_zWhA · 2023-11-22
> **Reply to authors**
>
> Great, please do include the new results regarding high-dimensional data/real-world data and the new discussion regarding other scalable/efficient alternatives in the final version. Also thanks for the new results regarding #proj, which are instructive.
>
> I will determine my final score after discussing with other reviewers and checking the final manuscript more carefully. As a general opinion, I think your work a quite fine contribution, and I will recommend acceptance.

---

> > ### Author Response · Authors · 2023-11-23
> >
> > Dear Reviewer zWhA,
> >
> > Thanks for your further response.
> >
> > We have updated our paper with the suggested/promised changes. Please post any other modifications that you see fit (or that we accidentally missed).
> >
> > Even though the discussion phase is closing soon, we will incorporate them in the camera-ready.
> >
> > Thanks again!

---

### Official Review · Reviewer_9kNo · 2023-10-30

**Soundness:** 2 fair
**Presentation:** 3 good
**Contribution:** 2 fair
**Rating:** 3
**Confidence:** 4

**Summary:**

The paper explores mutual information estimation utilizing a feedforward neural network. Unlike most methods based on the Donsker-Varadhan formula, such as MINE, there is no need to train a critic function. The author also introduces a feedforward network designed to estimate maximal correlation. These methods forego the need for per-distribution optimization, enhancing their efficiency.

**Strengths:**

The paper is well-written with a clear motivation.
The idea of replacing the optimization procedure with a feedforward neural network is intriguing.

**Weaknesses:**

1. The author fails to specify the dimension of the data $(x, y)$. If these are scalars, the task of mutual information estimation becomes considerably simpler. There exists an extensive body of literature on classical (non-neural network) methods addressing this, which warrants consideration.

2. Mutual information estimation methods based on the Donsker-Varadhan formula, such as MINE, are primarily designed for estimating mutual information between complex real-world data (e.g., CLIP). Comparing these to simple scalar setups seems inequitable. Should the proposed method be effective with high-dimensional data, the author is advised to validate it through experimentation.

3. Relying solely on Gaussian Mixture Models for training appears to be a limited approach, raising questions about its generalizability to other distributions. To validate training based solely on Gaussian Mixtures, the author should experiment with a variety of distributions. For further reference:
Czyz et al., "Beyond Normal: On the Evaluation of Mutual Information Estimators."

4. The core concept is to approximate the critic function $\theta$ which maximizes the Donsker-Varadhan formula using a neural network. The function that truly maximizes this is given by $\log p(x,y)/p(x)p(y)$. It seems implausible that a single pretrained neural network could accomplish this for every joint distribution $p(x, y)$.

**Questions:**

Please see Weaknesses section.

---

> ### Author Response · Authors · 2023-11-21
>
> Dear Reviewer 9kNo,
>
> Thank you for your constructive comments and insightful feedback on our manuscript. We highly appreciate your recognition of the paper's clarity, and that the idea of using a feedforward neural network for mutual information estimation is intriguing.
> We acknowledge your concerns regarding the dimensionality of the data, the comparison with classical methods, and the generalizability of our approach. We have addressed each of your points in our revised manuscript and believe that our additional experiments and clarifications will shed more light on the efficacy and applicability of our proposed method.
> We hope that these clarifications can help finalize your assessment and the rating of our paper. Please also let us know if you have any further questions that we need to provide additional clarifications.
>
> ***
>
> **W1: The author fails to specify the dimension of the data $(x,y)$. If these are scalars, the task of mutual information estimation becomes considerably simpler. There exists an extensive body of literature on classical (non-neural network) methods addressing this, which warrants consideration.**
>
> **A**: Thank you for highlighting the point of specifying the data dimensionality in mutual information estimation.
>
> Our work primarily focuses on scalar variables in classical mutual information (MI) estimation. However, it can be directly applied to estimating MI between high-dimensional random variables with the technique of sliced mutual information [1]. We have rigorously evaluated our InfoNet model on high-dimensional random variables, in terms of correlation order accuracy and independence test.
>
> Specifically, we validated the proposed InfoNet in two tasks:
>
> Task 1: given three jointly sampled random variables, X, Y, and Z from high-dimensional Gaussian, we want to know which one of Y and Z is more correlated with X, where the correct order, i.e., MI(X,Y)>=MI(X,Z) or MI(X,Y)<MI(X,Z)  is obtained by the ground-truth mutual information (MI) computed with the analytical solution of MI for Gaussian. The performance metric is the binary classification accuracy.
>
> Task 2: Independence test. Given a pair of random variables X, and Y, either independent or not, we apply InfoNet and MINE to check their independence since both metrics can claim independence between two random variables if the metrics are close to zero. We vary the epsilon as the threshold, and we plot the precision-recall curve to compare their performance using the area under the curve (AUC, the closer to 1, the better).
>
>
> Results:
>
> A. In terms of the correlation order test (Task 1), we get the following performance:
>
> | Dimensions | 2    | 3    | 4    | 5    | 6    | 7    | 8    | 9    | 10   |
> |------------|------|------|------|------|------|------|------|------|------|
> | MINE-5000  | 96.2 | 97   | 97   | 96.2 | 94.9 | 94.2 | 93.2 | 92.8 | 93   |
> | InfoNet | **97.7** | **96.4** | **96.2** | **97.9** | **97.4** | **98.1** | **98.2** | **97.3** | **98.3** |
>
> From the table, we can see that InfoNet performs better in terms of estimating the correlation orders between a triplet of random variables across multiple dimensions.
> Moreover, InfoNet runs at 0.05s per estimate, while MINE runs at 12.3s per estimate.
>
> Details of this experiment can be seen in B.3.1 of the revised version.
>
>
> B. In terms of the independence test, we have the AUCs of MINE-x (where x means the test-time gradient steps) and InfoNet obtained under different dimensions and sequence lengths (number for data points in each sequence):
>
> | Sequence length | 50    | 100    | 150    | 200    | 250    | 300    | 350    | 400
> |------------|------|------|------|------|------|------|------|------|
> | MINE-1000 d=16 | 0.757 | 0.668 | 0.843 | 0.902 | 0.996 | 1.0 | 1.0 | 1.0 | 1.0 |
> | MINE-1000 d=128 | 0.45 | 0.633 | 0.520 | 0.607 | 0.886 | 0.949 | 0.848 | 0.954 | 0.877 |
> | InfoNet d=16  | 0.914 | 0.992   | 1.0   | 1.0 | 1.0 | 1.0 | 0.99 | 1.0 |
> | InfoNet d=128  | 0.694 | 0.603   | 0.819   | 0.832 | 0.954 | 0.950 | 0.981 | 0.992 |
>
> From the Table, we can observe that the AUCs obtained with InfoNet are consistently higher than the ones obtained with MINE, showing that InfoNet provides a better estimation than MINE of mutual information despite a much shorter running time.
>
> Detailed plots and results can be found in B.3.2 of the revised version.
>
> Our findings, detailed in B.3 in the revised paper, demonstrate that InfoNet, leveraging sliced mutual information, exhibits robust performance at low time complexity in comparison to existing methods for estimating mutual information in high dimension.
>
> [1] Goldfeld Z, Greenewald K. Sliced mutual information: A scalable measure of statistical dependence[J]. Advances in Neural Information Processing Systems, 2021, 34: 17567-17578.

---

> > ### Author Response · Authors · 2023-11-21
> > **Replying to Official Comment by Authors**
> >
> > **W2: "Mutual information estimation methods based on the Donsker-Varadhan formula, such as MINE, are primarily designed for estimating mutual information between complex real-world data (e.g., CLIP). Comparing these to simple scalar setups seems inequitable. Should the proposed method be effective with high-dimensional data, the author is advised to validate it through experimentation."**
> >
> > **A**: Thank you for pointing out this. We have included more results on high-dimensional data, as discussed in W1.
> >
> > Besides, we add another evaluation task to show that the InfoNet model is capable of handling high-dimensional real data. The task is the correlation order prediction, which means, given three random variables, X,Y, and Z, we would like to find which one of Y and Z is more correlated with X. However, it is performed with high-dimensional features from real images.
> >
> > Specifically, we download the ImageNet2017(ILSVRC) dataset and CLIP pre-trained model. Denote the image encoder of CLIP as a function $f_{CLIP}$, which takes in an image and outputs a 512-dimension vector representing the feature of the image. We begin with a randomly selected sequence of images from ILSVRC dataset. Then $X$ is generated by $X=f_{CLIP}(image)$ to be a sequence of 512-dimensional vectors. Then we add Gaussian noise to the original image to get $Y$: $Y = f_{CLIP}(image + noise)$. Finally, we add color jitter to randomly change the image's brightness, contrast, saturation, and hue. We denote the transformation as $g_{jitter}$. Then $Z$ is generated by $Z=f_{CLIP}(g_{jitter}(image+noise))$. In this case, we can tentatively ensure that the ground truth is $MI(X,Y)>MI(X,Z)$ as the information contained in the image for getting Z is less than that for getting Y with respect to the original image, according to the Data Processing Inequality.
> >
> >
> > We apply our InfoNet model on this task using sliced mutual information to test the order and reach 100% accuracy. It shows that our method can generalize to real-world data in high dimensions. The accuracy comparison is listed below:
> >
> > | Method | Ours | MINE |
> > |------------|----|----|
> > | accuracy  | 100 | 100 |
> >
> > ***
> >
> > **W3: ``Relying solely on Gaussian Mixture Models for training appears to be a limited approach, raising questions about its generalizability to other distributions. To validate training based solely on Gaussian Mixtures, the author should experiment with a variety of distributions. For further reference: Czyz et al., "Beyond Normal: On the Evaluation of Mutual Information Estimators."'**
> >
> > **A**: We appreciate your observation regarding the scope of our experiments. Following your suggestion, we have now included tests on three different one-dimensional distributions from the reference, each with known ground truths for mutual information. Our InfoNet model, despite being trained solely on GMMs, has shown promising results across these additional distributions, underscoring its promising generalizability. For a thorough examination of these results, we invite you to review the updated section B.1.1 in the revised paper, illustrated in Figure 15.
> >
> > Moreover, in our revision, we have included a copula technique. We add it as a data preprocessing step before the original data is put into the network. Using the copula technique to transfer the marginal distribution of $(X,Y)$ to uniform distribution between [0,1] makes it much easier to learn. Not only could the network reach much better performance, but also it sped up the convergence of the training by 5 times. The newest results of the order test and the error mean/variance of the MI estimate are shown in Section B.1 of the revised paper, in Tables 6 and 7 respectively. We also present the estimated mutual information with ground truth MI under the different number of Gaussian components (See Figure 14). From the figure, we can see that our method is very close to the ground truth.

---

> > > ### Author Response · Authors · 2023-11-21
> > >
> > > **W4: ``The core concept is to approximate the critic function $\theta$ which maximizes the Donsker-Varadhan formula using a neural network. The function that truly maximizes this is given by $\log p(x,y)/p(x)p(y)$. It seems implausible that a single pretrained neural network could accomplish this for every joint distribution $p(x,y)$.''**
> > >
> > > **A**: We fully acknowledge the concern raised about the capacity of a single pre-trained neural network to approximate the critic function \( \theta \) across all joint distributions \( p(x,y) \).
> > >
> > > However, we would like to argue that due to the fact that any distribution can be faithfully approximated by a mixture of Gaussian (MoG) with sufficient capacity, the proposed training of the InfoNet, in theory, should endow it with reasonable generalization capability.
> > >
> > > Our experiments also support this claim. For example, we have performed two additional experiments in W1, where the trained InfoNet is directly applied to high-dimensional random variables and demonstrates sound performance. Also, we have applied the pretrained InfoNet on various distributions apart from Gaussian in W3, and show MI estimates close to ground truth.
> > >
> > > Actually, we were also supervised when seeing that the proposed InfoNet can overfit (learn to optimize) to the training distributions, given that we are sampling infinitely many MoGs from a wide range of MoG parameters. But this happened due to the network design, e.g., the proposed attention mechanism and the look-up table formula. Also, drawing parallels with large-scale models like GPT, which exhibit remarkable capabilities given extensive data and computational power, we believe that such an outcome is within the realm of possibility for neural networks. Although our current training is confined to Gaussian Mixture Models due to data and computational constraints, we are optimistic about the potential to significantly enhance InfoNet's generalization by incorporating diverse data in the future.
> > >
> > > Last but not least, as an initial attempt to eliminate the computation overhead of MI (especially during test time) using a feed-forward neural network architecture, we believe that the proposed method and the experiments shall provide evidence to the community that we may have an alternative to performing highly efficient mutual information estimation on a wide range of distributions in high dimensional space.

---

### Official Review · Reviewer_FUUK · 2023-11-01

**Soundness:** 3 good
**Presentation:** 4 excellent
**Contribution:** 3 good
**Rating:** 8
**Confidence:** 4

**Summary:**

In this paper, the authors have proposed a neural network approach using the attention-based mechanism called InfoNet for mutual information estimation. Using the Donsker & Varadhan representation of the KL divergence, the parameter of the network can be optimized by maximizing an objective function (Equation 2). Furthermore, an evaluation of the efficacy of the proposed approach on the Hirschfeld-Gebelein-R´enyi (HGR) Maximum Correlation Coefficient has been considered.

**Strengths:**

Speeding up the previous similar work, MINE by Belghazietal., 2018 using highly parallelizable attention modules.

**Weaknesses:**

Overall the paper is well written, and I don't have any specific concerns. I just wanna clarify two things:

1- Is speeding up MINE using the attention architecture the only difference between InfoNet and MINE? Fundamentally, these two algorithms are similar and the same objective function using the same Donsker & Varadhan representation of the KL divergence has been used for the MI estimation.

2 - While the authors have mentioned some motivations for evaluating the proposed approach for MMC, there are other dependence measures with computational efficiency that also enjoy the properties of MI. For instance, Energy Distance (ED) by Gábor  Székely is one of them which is a dual form of Maximum Mean Discrepancy (MMD). In particular, ED  does not require any parametric assumption about the underlying probability distribution of the data, making it easy for many statistical inference problems. It is a good idea to compare the proposed MI estimator and MCC with ED in terms of the correlation of two random variables (similar to Figure 4 ) and computational speed given the fact that ED is quite fast to use in high dimensional data.

**Questions:**

Please see my above comments.

---

> ### Author Response · Authors · 2023-11-21
>
> Dear Reviewer FUUK,
>
> Thank you for your insightful and constructive review of our manuscript. We greatly appreciate the time and effort you have devoted to evaluating our work, and your positive remarks on our proposed feed-forward neural network approach, InfoNet, using the attention-based mechanism for mutual information estimation, as well as the excellent presentation of the work.
>
> We have taken a detailed look into your questions and comments, and have prepared comprehensive responses to each of your points, which we hope will address the issues effectively. We followed your suggestion to explore the comparison with other dependence measures like Energy Distance.
>
> We hope that these clarifications can help finalize your assessment and the rating of our paper. Please also let us know if you have any further questions that we need to provide additional clarifications.
>
> ***
>
> **W1 Is speeding up MINE using the attention architecture the only difference between InfoNet and MINE? Fundamentally, these two algorithms are similar and the same objective function using the same Donsker \& Varadhan representation of the KL divergence has been used for the MI estimation.**
>
> **A**: Thanks for your comment. Speeding up MINE with the attention architecture is not the only difference. To elaborate, we first provide a brief summary of MINE and InfoNet.
>
> *MINE*: optimizes a neural network (NN) with the Donsker-Varadhan (DV) objective for a sequence \{$x_t,y_t$\} ($t=1,..., T$) sampled from a single distribution $p(x,y)$. When the optimization is done, the mutual information (MI) estimation of $p(x,y)$ is obtained. If one needs to estimate the MI of a different distribution $p’(x,y)$ using MINE, the neural network will be optimized again using the DV objective, but the sequence sampled from $p’(x,y)$. Since the optimal parameters of an NN ($\theta^*$) are different for different distributions.
>
> *InfoNet*: optimizes a neural network (the proposed) also with the DV objective, but for many different distributions, i.e., sequences \{$x_t^i,y_t^i$\} ($t=1,..., T$,$i=1,..., N$) are sampled from N different distributions $p^i(x,y)$’s. When the optimization is done, it can be used to estimate the MI of an arbitrary distribution via the generalization capability of the proposed InfoNet architecture, without re-training. Equivalently, InfoNet, with a sequence as input, can directly output the optimal set of parameters ($\theta^*$) corresponding to the one obtained by performing a MINE optimization for a distribution (without finetuning InfoNet on this distribution).
>
> With the above in mind, the key differences between MINE and InfoNet are listed below:
>
> 1. During training, MINE optimizes the NN for a single distribution, but InfoNet optimizes the proposed NN for many distributions.
>
> 2. During training, MINE treats the NN as a (scalar) discriminant whose optimum (defined by the DV formula and the sequence) gives the MI of the distribution represented by the input sequence. However, InfoNet acts as a hyper-network, and InfoNet outputs the optimal discriminant (NN in MINE) corresponding to an arbitrary distribution from the input sequence.
>
> 3. During testing, when a new distribution arrives, MINE needs to optimize the NN for estimating its distribution with the sequence sampled from this new distribution. In contrast, InfoNet takes the sampled sequence from this new distribution, and directly outputs the optimal NN (represented as a look-up table) as well as the MI estimation without performing any network update.

---

> ### Author Response · Authors · 2023-11-21
> **Replying to Official Comment by Authors**
>
> **W2: It is a good idea to compare the proposed MI estimator and MCC with ED in terms of the correlation of two random variables (similar to Figure 4 ) and computational speed given the fact that ED is quite fast to use in high dimensional data.**
>
>
> **A**: Thank you for bringing up the comparison with Energy Distance (ED). Indeed, ED is a noteworthy non-parametric measure known for its computational efficiency, especially in high-dimensional data scenarios, and it is a good measure in measuring the differences between two distributions.
>
> We follow the suggestion and compare InfoNet (operated under 1000 random slices similar to [1]) with ED [2] in terms of computational efficiency and performance. Specifically, we consider two tasks.
>
> Task 1: given three jointly sampled random variables, X, Y, and Z from high-dimensional Gaussian, we want to know which one of Y and Z is more correlated with X, where the correct order, i.e., MI(X,Y)>=MI(X,Z) or MI(X,Y)<MI(X,Z)  is obtained by the ground-truth mutual information (MI) computed with the analytical solution of MI for Gaussian. The performance metric is the binary classification accuracy.
>
> Task 2: Independence test. Given a pair of random variables X, and Y, either independent or not, we apply InfoNet and ED to check their independence since both metrics can claim independence between two random variables. We vary the epsilon as the threshold, and we plot the precision-recall curve to compare their performance using the area under the curve (AUC, the closer to 1, the better).
>
> Results:
>
> A. Efficiency: Time compare:
> | Dimension    | 16 | 128 |
> |------------|----|----|
> | ED  | 0.035 | 0.224 |
> | Ours  | 0.050 | 0.058 |
>
> From the Table, we can see that the efficiency of InfoNet is not sensitive to the dimension of the random variable, and InfoNet is more efficient than ED at a higher dimension.
>
> B. In terms of the correlation order test (Task 1), we get the following performance:
>
> | Dimensions | 2    | 3    | 4    | 5    | 6    | 7    | 8    | 9    | 10   |
> |------------|------|------|------|------|------|------|------|------|------|
> | Energy Distance | 49.6 | 51.2 | 52.2 | 51.5 | 52.5 | 49.6 | 48.7 | 50.2 | 51.3 |
> | MINE-5000  | 96.2 | 97   | 97   | 96.2 | 94.9 | 94.2 | 93.2 | 92.8 | 93   |
> | InfoNet | **97.7** | **96.4** | **96.2** | **97.9** | **97.4** | **98.1** | **98.2** | **97.3** | **98.3** |
>
> From the table, we can see that InfoNet performs better in terms of estimating the correlation orders between a triplet of random variables across multiple dimensions.
>
> Details of this experiment can be seen in B.3.1 of the revised version.
>
> C. In terms of the independence test, we have the AUCs of ED and InfoNet obtained under different dimensions and sequence lengths:
>
> | Sequence length | 50    | 100    | 150    | 200    | 250    | 300    | 350    | 400 |
> |------------|------|------|------|------|------|------|------|------|
> | Energy Distance d=16 | 0.733 | 0.964 | 1.0 | 1.0 | 1.0 | 1.0 | 1.0 | 1.0 |
> | Energy Distance d=128 | 0.167 | 0.1 | 0.265 | 0.416 | 0.473 | 0.768 | 0.879 | 0.981 |
> | InfoNet d=16  | 0.914 | 0.992   | 1.0   | 1.0 | 1.0 | 1.0 | 0.99 | 1.0 |
> | InfoNet d=128  | 0.694 | 0.603   | 0.819   | 0.832 | 0.954 | 0.950 | 0.981 | 0.992 |
>
> From the Table, we can observe that the AUCs obtained with InfoNet are consistently higher than the ones obtained with ED, showing that InfoNet provides a better metric than ED.
>
> Detailed plots and results can be found in B.3.2 of the revised version.
>
>
> [1] Goldfeld Z, Greenewald K. Sliced mutual information: A scalable measure of statistical dependence[J]. Advances in Neural Information Processing Systems, 2021, 34: 17567-17578.
>
> [2] Rizzo M L, Székely G J. Energy distance[J]. wiley interdisciplinary reviews: Computational statistics, 2016, 8(1): 27-38.

---

> ### Comment · Reviewer_FUUK · 2023-11-23
>
> Thanks for your effort to include more clarification about your approach in comparison with MINE, and adding more experiments about ED. I have read the comments and the revised version and increased my score from 6 to 8 (There is no 7 in the scores :)).

---

### Official Review · Reviewer_C35n · 2023-11-02

**Soundness:** 2 fair
**Presentation:** 2 fair
**Contribution:** 2 fair
**Rating:** 6
**Confidence:** 4

**Summary:**

This paper investigates the estimation of mutual information between two random variables, primarily focusing on sequences, by leveraging a neural network-based approach for efficiency. To be specific, the authors introduce the application of an attention mechanism for feed-forward predictions, trained with the MINE-based training objective. Experimental results demonstrate the effectiveness of the proposed approach with different families of distribution, as well as its promising results on real-world applications.

**Strengths:**

### Originality

The paper builds on two existing works: MINE [Belghazi et al.] and attention mechanism [Jaegle et al.]. The former is used as a training objective, while the latter is used to parameterize the function in MINE. The combination of two existing ideas enables efficient computation for mutual information between two sequences.


### Significance

Estimating mutual information between two high-dimensional variables is important, yet challenging for many real-world tasks. The paper proposes one way to efficiently compute MI between two sequences.

**Weaknesses:**

I am confused with some parts, might be due to writing or organization. Please see the following questions.

**Questions:**

### Introduction

- 'Specifically, we want to explore whether the estimation of mutual information can be performed by a feed-forward prediction of a neural network' -- What does a feed-forward prediction mean? For MINE, we still use NNs to parameterize a function and output a scalar via NNs. Is MINE a feed-forward prediction? Please elaborate it.

- 'Moreover, each time the joint distribution changes (different sequences), a new optimization has to be performed, thus not efficient.' -- For Figure 1, which type of sequences are you considering? I don't understand 'a new optimization has to be performed'. Could you please elaborate more? Figure 1 lacks necessary contexts.

- 'This way, we transform the optimization-based estimation into a feed-forward prediction, thus bypassing the time-consuming gradient computation and avoiding sub-optimality via large-scale training on a wide spectrum of distributions.' -- For MINE, we do need to update NNs' parameters. But InfoNet also needs gradient ascent. How to understand 'bypassing the time-consuming gradient computation'?

All in all, I think the confusion is due to the lack of enough explanations. Could you please elaborate the difference between MINE and InfoNet? The same training objective is used. Both can parameterize the function via NNs, even the same NN architecture.


### Method

- To estimate mutual information between two sequences in Eq.(2), InfoNet only considers $(x_{t}, y_{t})$ at each time step? Thus, it ignores mutual information between e.g., $x_{1}$ and $y_{2}$?

- In terms of Figure 2, the difference between MINE and InfoNet is just a look-up table? I think same NN architectures could also be used in MINE?

---
**Update after rebuttal**: I raised the score: 5 --> 6.

---

> ### Author Response · Authors · 2023-11-21
>
> Dear Reviewer C35n,
>
> Thank you for your careful review and constructive questions. We also appreciate your acknowledgment of the importance of the problem we are tackling, and the efficiency of the InfoNet we are proposing.
>
> We understand that there exists a misunderstanding, and the key differences between MINE and InfoNet need to be clarified. Accordingly, we have revised the paper with all the clarifications derived from these questions, which are answered in detail below.
>
> We hope that these clarifications can help finalize your assessment and the rating of our paper. Please also let us know if you have any further questions that we need to provide additional clarifications.
>
> ***
>
> **W1: I am confused with some parts, might be due to writing or organization. Please see the following questions.**
>
> **A**: Thanks for all your questions below. We understand that there is confusion on some parts due to the lack of enough explanations on the key differences between MINE and InfoNet. Next, will first elaborate on the key differences, and then provide clarifications for each of the questions.
>
> ***
>
> **Q1: All in all, I think the confusion is due to the lack of enough explanations. Could you please elaborate on the difference between MINE and InfoNet? The same training objective is used. Both can parameterize the function via NNs, even the same NN architecture.**
>
> **A**: First, we provide a brief summary of MINE and InfoNet.
>
> *MINE*: optimizes a neural network (NN) with the Donsker-Varadhan (DV) objective for a sequence \{$x_t,y_t$\} ($t=1,..., T$) sampled from a single distribution $p(x,y)$. When the optimization is done, the mutual information (MI) estimation of $p(x,y)$ is obtained. If one needs to estimate the MI of a different distribution $p’(x,y)$ using MINE, the neural network will be optimized again using the DV objective, but the sequence sampled from $p’(x,y)$. Since the optimal parameters of an NN ($\theta^*$) are different for different distributions.
>
> *InfoNet*: optimizes a neural network (the proposed) also with the DV objective, but for many different distributions, i.e., sequences \{$x_t^i,y_t^i$\} ($t=1,..., T$,$i=1,..., N$) are sampled from N different distributions $p^i(x,y)$’s. When the optimization is done, it can be used to estimate the MI of an arbitrary distribution via the generalization capability of the proposed InfoNet architecture, without re-training. Equivalently, InfoNet, with a sequence as input, can directly output the optimal set of parameters ($\theta^*$) corresponding to the one obtained by performing a MINE optimization for a distribution (without finetuning InfoNet on this distribution).
>
> With the above in mind, the key differences between MINE and InfoNet are listed below:
>
> 1. During training, MINE optimizes the NN for a single distribution, but InfoNet optimizes the proposed NN for many distributions.
>
> 2. During training, MINE treats the NN as a (scalar) discriminant whose optimum (defined by the DV formula and the sequence) gives the MI of the distribution represented by the input sequence. However, InfoNet acts as a hyper-network, and InfoNet outputs the optimal discriminant (NN in MINE) corresponding to an arbitrary distribution represented by the input sequence.
>
> 3. During testing, when a new distribution arrives, MINE needs to optimize the NN for estimating its MI with the sequence sampled from this new distribution. In contrast, InfoNet takes the sampled sequence from this new distribution, and directly outputs the optimal NN (represented as a look-up table) as well as the MI estimation without performing any network update.
>
> ***
>
> **Q2: 'Specifically, we want to explore whether the estimation of mutual information can be performed by a feed-forward prediction of a neural network' -- What does a feed-forward prediction mean? For MINE, we still use NNs to parameterize a function and output a scalar via NNs. Is MINE a feed-forward prediction? Please elaborate it.**
>
> **A**: Thanks for the question. MINE optimizes the NN, which outputs a scalar for a data point in the sequence and accumulates the scalars to get the MI estimate of the distribution represented by the sequence. However, MINE has to perform another optimization for a different distribution (or sequence), with which, we state that MINE is not feed-forward. In other words, MINE is feed-forward for predicting the scalar for a data point, but not feed-forward for estimating the MI of a distribution.
>
> In contrast, after massive training on a spectrum of distributions, when a new distribution is presented to InfoNet in the form of a sequence, InfoNet predicts the optimal discriminant (without changing its parameters), which then computes the scalars and the MI estimate of the new distribution. So, InfoNet is feed-forward for computing the MI of a distribution, no need to perform training for a new distribution. Thus InfoNet is much more efficient than MINE as no gradient ascent is needed with a test distribution.

---

> > ### Author Response · Authors · 2023-11-21
> > **Replying to Official Comment by Authors**
> >
> > ***
> >
> > **Q3: 'Moreover, each time the joint distribution changes (different sequences), a new optimization has to be performed, thus not efficient. ' -- For Figure 1, which type of sequences are you considering? I don't understand why a new optimization has to be performed. Could you please elaborate more? Figure 1 lacks the necessary contexts.**
> >
> > **A**: For training InfoNet, we first randomly sample a mixture of Gaussian (MoG, from a wide range to ensure generalization), and then sample a sequence from this MoG. The whole sequence is input to InfoNet to compute the optimal discriminant (note the NN discriminant in MINE is used to compute a scalar from a data point from this sequence).
> >
> > For MINE, the NN is trained for a single distribution or sequence, so it can not be used to compute the MI for a new sequence. And to do so, MINE has to perform a new optimization of the NN when facing a new distribution. In contrast, InfoNet does not need to perform optimization to get the optimal discriminant for a new distribution, but just needs to perform a feed-forward pass with the new sequence (distribution) as input, which is much more efficient.
> >
> > ***
> >
> > **Q4: 'This way, we transform the optimization-based estimation into a feed-forward prediction, thus bypassing the time-consuming gradient computation and avoiding sub-optimality via large-scale training on a wide spectrum of distributions. ' -- For MINE, we do need to update NNs' parameters. But InfoNet also needs gradient ascent. How to understand 'bypassing the time-consuming gradient computation'?**
> >
> > **A**: Thanks for the question. Yes, during training, both MINE and InfoNet need gradient ascent. However, during testing for a different distribution, MINE still needs gradient ascent. While InfoNet can predict the optimal discriminant via an efficient feed-forward pass without any gradient ascent. So InfoNet eliminates test-time time-consuming gradient computation. We have revised the draft.
> >
> > ***
> >
> > **Q5: To estimate mutual information between two sequences in Eq.(2), InfoNet only considers $(x_t,y_t)$ at each time step? Thus, it ignores mutual information between e.g., $x_1$ and $y_2$?**
> >
> > **A**: InfoNet takes the whole sequence \{$x_t,y_t$\} ($t=1,..., T$)  as input, and outputs the optimal discriminant, which can be queried to efficiently compute the MI of the distribution that synthesizes  \{$x_t,y_t$\} ($t=1,..., T$). Thanks for the question; we have made the update.
> >
> > ***
> >
> > **Q6: In terms of Figure 2, the difference between MINE and InfoNet is just a look-up table? I think same NN architectures could also be used in MINE?**
> >
> > **A**: The major differences are listed in Q1. And the look-up table is one key difference that ensures that the proposed InfoNet can be trained for performing the MI estimation of many different distributions without test-time gradient ascent. The other key differences are elaborated in Q1. Please refer to it for more details.

---

> > ### Comment · Reviewer_C35n · 2023-11-22
> > **Thank you for the response!**
> >
> > Thank you for the detailed response. I appreciate all the efforts to improve the paper and the new experiments!
> >
> > The key differences between MINE and InfoNet are clear to me now. MINE is tailored for a specific data distribution. When a new data distribution comes in, it might fail without being re-trained on the new distribution. On the other hand, InfoNet looks like a universal mutual information neural estimator. Thus, it seems similar to what pre-trained models are doing.
> >
> > Additionally, the DV objective is essentially similar to the training objective used for contrastive learning, where pre-training has gained popularity, even though these approaches operate in different domains.
> >
> > I raised the score: 5 --> 6.

---

> > > ### Author Response · Authors · 2023-11-22
> > >
> > > Dear Reviewer C35n,
> > >
> > > Thanks very much for your positive reception of our endeavor to improve the paper. Also, thanks for your recognition of the key differences between MINE and InfoNet. Your insight on the relationship between DV and contrastive learning is inspiring, and we will incorporate a discussion on this in our final version.
> > >
> > > Thanks again for elevating the rating of our paper.

---

### Author Response · Authors · 2023-11-21

We sincerely thank all reviewers for their valuable comments and constructive suggestions. We appreciate the positive responses from the reviewers on several aspects of our work: 1) Significance (R1, R2, R4): Addresses the fundamental problem of scalable estimation of mutual information and statistical dependence in data science and machine learning; combines the accuracy of neural network-based methods with the efficiency of non-parametric methods. 2) Novelty (R1, R3, R4): Presents a novel approach within the context of MI and statistical dependence estimation; offers an orthogonal possibility to recent works; first to apply simulation-based learning to statistical dependence modeling. And 3) Presentation (R2, R3, R4): The paper is well-written and easy to follow, providing an enjoyable reading experience.

We have performed **additional 4 experiments (3 multi-dimension and 1 one-dimension) and revised the manuscript accordingly**, which reflects the reviewers' comments (all revisions are highlighted in blue color in the new PDF). The updates are summarized as follows:

**Appendix B.1**: Introduce the copula technique we use and present the latest results from incorporating the copula.

**Appendix B.2**: We represent the experiments checking the performance on one-dimensional distributions in addition to the Mixture of Gaussian. (R3)

**Appendix B.3**: Introduce the sliced mutual information technique for estimation of high dimensional data.

**Appendix B.3.1**: Show the experiment result of correlation order for multi-dimensional gauss distributions (R2, R3, R4).

**Appendix B.3.2**: Show the experiment result for high-dimensional independence testing (R2, R3, R4).

**Appendix B.3.3**: Show the experiment result of correlation order using data generated by CLIP image encoder (R3).

**Appendix B.4**: We add a discussion on the relationship between our work and simulation-based intelligence (R4).

**Section 3**: We separate sections 3.1 and 3.2 into a new section called Training Algorithm (R4).

We appreciate the valuable insights and contributions from the reviewers to improving our manuscript. We have carefully addressed each query and point raised in the following, and are grateful for any further clarification needed to advance our score.

Once again, thank you for your thoughtful review!

---

### Meta-Review · Area_Chair_qiBH · 2023-12-10

**Metareview:**

The reviewers raised multiple concerns, including the experimental setting, efficiency and writing. Despite likely improvements from author feedback, a consensus on paper acceptance wasn't reached due to a brief discussion period and substantial revisions undertaken during this period.

**Justification For Why Not Higher Score:**

Despite likely improvements from author feedback, a consensus on paper acceptance wasn't reached due to a brief discussion period and substantial revisions undertaken during this period.

**Justification For Why Not Lower Score:**

n/a

---

### Decision · Program_Chairs · 2024-01-16

Reject